# Rogation ceremonies: A key to understanding past drought variability in north-eastern Spain since 1650

*Tejedor E[1,2], de Luis M [1,2], Barriendos M[3], Cuadrat JM[1,2], Luterbacher J[4,5], Saz MA[1,2]

[1]Dept. of Geography and Regional Planning. University of Zaragoza. Zaragoza. (Spain).

[2]Environmental Sciences Institute of the University of Zaragoza. Zaragoza. (Spain).

[3]Department of History. University of Barcelona (Spain).

[4]Department of Geography, Climatology, Climate Dynamics and Climate Change, Justus Liebig University Giessen, Germany

[5]Centre for International Development and Environmental Research, Justus Liebig University Giessen, Germany

*Correspondence to: Miguel Ángel Saz; masaz@unizar.es

**ABSTRACT**

In the northeast of the Iberian Peninsula, few studies have reconstructed drought occurrence and variability for the pre-instrumental period using documentary evidence and natural proxies. In this study, we compiled a unique dataset of rogation ceremonies - religious acts asking God for rain - from 13 cities in the north-east of Spain and investigated the annual drought variability from 1650 to 1899 AD. Three regionally different coherent areas (Mediterranean, Ebro Valley and Mountain) were detected. Both the Barcelona and the regional Mediterranean drought indices were compared with the instrumental series of Barcelona for the overlapping period (1787-1899), where we discovered a highly significant and stable correlation with the Standardized Precipitation Index of May with a 4-month lag ($r=-0.46$ and $r=-0.53$; $p<0.001$, respectively). We found common periods with prolonged droughts (during the mid and late 18th century) and extreme drought years (1775, 1798, 1753, 1691 and 1817) associated with more atmospheric blocking situations. A superposed epoch analysis (SEA) was performed showing a significant decrease in drought events one year after the volcanic events, which might be explained by the decrease in evapotranspiration due to reduction in surface temperatures and, consequently, the higher availability of water that increases soil moisture. In addition, we discovered a common and significant drought response in the three regional drought indices two years after the Tambora volcanic eruption. Our study suggests that documented information on rogations contains important independent evidence to reconstruct extreme drought events in areas and periods for which instrumental information and other proxies are scarce. However, drought index at Mountain areas presents various limitations and its interpretation must be treated with caution.

## 1. Introduction

Water availability is one of the most critical factors for human activities, human wellbeing and the sustainability of natural ecosystems. Drought is an expression of a

precipitation deficit, which often lasts longer than a season, a year or even a decade.
Drought leads to water shortages associated with adverse impacts on natural systems
and socioeconomic activities, such as reductions in streamflow, crop failures, forest
decay or restrictions on urban and irrigation water supplies (Eslamian and Eslamian,
2017). Droughts represent a regular, recurrent process that occurs in almost all climate
zones. In the Mediterranean region, the impacts of climate change on water resources
give significant cause for concern. Spain is one of the European countries with a large
risk of drought caused by high temporal and spatial variability in the distribution of
precipitation (Vicente-Serrano et al., 2014; Serrano-Notivoli et al., 2017). Several recent
Iberian droughts and their impacts on society and the environment have been
documented in the scientific literature (e.g., Dominguez Castro et al., 2012; Trigo et al.
2013; Vicente-Serrano et al. 2014; Russo et al. 2015; Turco et al. 2017). For instance,
during the period from 1990 to 1995, almost 12 million people suffered from water
scarcity, the loss in agricultural production was an estimated 1 billion Euro, hydroelectric
production dropped by 14.5 % and 63% of southern Spain was affected by fires
(Dominguez Castro et al., 2012). One of the most recent droughts in Spain lasted from
2004 to 2005 (García-Herrera et al., 2007) and was associated with major socioeconomic
impacts (hydroelectricity and cereal production decreased to 40% and 60%,
respectively, of the average value).
In other European regions, drought intensity and frequency have been widely
studied, since their socio-economic and environmental impacts are expected to worsen
with climate change (e.g. Spinoni et al., 2018; Hanel et al., 2018). Long-term studies
using instrumental meteorological observations have helped in understanding European
drought patterns at various spatial and temporal scales (e.g. Spinoni et al., 2015; Stagge
et al., 2017). In addition, natural proxy data have provided a multi-centennial long-term
perspective in Europe by developing high-resolution drought indices derived mostly
from tree-ring records (e.g. Büntgen et al., 2010, 2011; Cook et al., 2015; Dobrovolný et
al. 2018). Finally, documentary records utilized in historical climatology have
complemented the understanding of droughts across Europe (e.g. Brázdil et al., 2005,
2010, 2018). These studies, covering the last few centuries, usually focus on specific
periods of extreme droughts and their societal impacts (e.g. Diodato and Bellochi, 2011;
Domínguez-Castro et al., 2012) and yet, studies attempting to develop continuous
drought indices for the last few centuries, inferred from documentary evidence, remain
an exception (e.g. Brázdil et al., 2013, 2016, 2018, 2019; Dobrovolný et al., 2015a,b;
Možný et al., 2016; Mikšovský et al., 2019 ).
In the Iberian Peninsula, natural archives including tree-ring chronologies, lake
sediments and speleothems have been used to deduce drought variability before the
instrumental period (Esper et al., 2015; Tejedor et al., 2016, 2017c; Benito et al., 2003,
2008; Pauling et al. 2006; Brewer et al., 2008; Carro-Calvo et al., 2013, Abrantes et al.,
2017, Andreu-Hayles et al., 2017). Nevertheless, most of the highly temporally resolved
natural proxy-based reconstructions represent high-elevation conditions during specific
periods of the year (mainly summer e.g. Tejedor et al., 2017c). Spain has a large amount
of documentary-based data with a good degree of continuity and homogeneity for many
areas, which enables important paleo climate information to be derived at different
timescales and for various territories. Garcia-Herrera et al. (2003) describe the main

archives and discuss the techniques and strategies used to derive climate-relevant information from documentary records. Past drought and precipitation patterns have been inferred by exploring mainly rogation ceremonies and historical records from Catalonia (Martin-Vide and Barriendos 1995; Barriendos, 1997; Barriendos and Llasat, 2003; Trigo et al. 2009), Zaragoza (Vicente-Serrano and Cuadrat, 2007), Andalusia (Rodrigo et al., 1998; 2000), central Spain (Domínguez-Castro et al., 2008; 2012; 2014; 2016) and Portugal (Alcoforado et al. 2000). In north-eastern Spain, the most important cities were located on the riverbanks of the Ebro Valley, which were surrounded by large areas of cropland (Fig. 1). Bad wheat and barley harvests triggered socio-economic impacts, including the impoverishment or malnutrition of whole families, severe alteration of the market economy, social and political conflicts, marginality, loss of population due to emigration and starvation, and diseases and epidemics, such as those caused by pests (Tejedor, 2017a). Recent studies have related precipitation/drought variability in regions of Spain to wheat yield variability (Ray et al., 2015; Esper et al. 2017). The extent of impacts caused by droughts depends on the socio-environmental vulnerability of an area, and is related to the nature and magnitude of the drought and the structure of societies, such as agricultural-based societies including trades (Scandyln et al., 2010; Esper et al. 2017).

During the past few centuries, Spanish society has been strongly influenced by the Catholic Church. Parishioners firmly believed in the will of God and the church to provide them with better harvests. They asked God to stop or provide rain through rogations, a process created by bishop Mamertus in AD 469 (Fierro, 1991). The key factor in evaluating rogation ceremonies for paleo-climate research is determining the severity and duration of adverse climatic phenomena based on the type of liturgical act that was organized after deliberation and decision-making by local city councils (Barriendos, 2005). Rogations are solemn petitions by believers asking God to grant specific requests (Barriendos 1996, 1997). Then, *pro-pluviam* rogations were conducted to ask for precipitation during a drought, and they therefore provide an indication of drought episodes and clearly identify climatic anomalies and the duration and severity of the event (Martín-Vide & Barriendos, 1995; Barriendos, 2005). In contrast, *pro-serenitate* rogations were requests for precipitation to end during periods of excessive or persistent rain causing crop failures and floods. In the Mediterranean basin, the loss of crops triggered severe socio-economic problems and was related to insufficient rainfall. Rogations were an institutional mechanism to address social stress in response to climatic anomalies or meteorological extremes (e.g. Barriendos, 2005). The municipal and ecclesiastical authorities involved in the rogation process guaranteed the reliability of the ceremony and maintained a continuous documentary record of all rogations. The duration and severity of natural phenomena that stressed society is reflected in the different levels of liturgical ceremonies that were applied (e.g. Martin-Vide and Barriendos, 1995; Barriendos, 1997; 2005). Through these studies, we learned that the present heterogeneity of drought patterns in Spain also occurred over the past few centuries, in terms of the spatial differences, severity and duration of the events (Martin-Vide, 2001, Vicente-Serrano 2006b). Nevertheless, the fact that no compilation

has been made of the main historical document datasets assembled over the past
several years is impeding the creation of a continuous record of drought recurrences
and intensities in the north-east of the Iberian Peninsula.

Here we compiled 13 series of historical documentary information of the *pro-*
*pluviam* rogation data from the Ebro Valley and the Mediterranean Coast of Catalonia
(Fig. 1) from 1438 to 1945 (Tab. 1). The cities cover a wide range of elevations from
Barcelona, which is near the sea (9 m a.s.l.), to Teruel (915 m a.s.l.) (Fig 1). Although
some periods have already been analyzed for certain cities (i.e., Zaragoza in 1600-1900
AD by Vicente-Serrano and Cuadrat, 2007; Zaragoza, Calahorra, Teruel, Vic, Cervera
Girona, Barcelona, Tarragona and Tortosa in 1750-1850 AD by Dominguez-Castro et al.,
2012; La Seu d'Urgell, Girona, Barcelona, Tarragona, Tortosa and Cervera in 1760-1800
AD by Barriendos and Llasat, 2003), this is the first systematic approach that analyzes all
existing information for north-eastern Spain, including new, unpublished data for
Huesca (1557-1860 AD) and Barbastro (1646-1925 AD) and examines the 13 sites jointly
over a period of 250 years (1650-1899 AD). We analyzed droughts across the sites and
identified extreme drought years and common periods in frequency and intensity. We
also analyzed statistical links between drought indices and major tropical volcanic
events in order to determine the effects of strong eruptions on regional droughts.

## 2. Methods
### 2.1.     Study area

The study area comprises the north-eastern part of Spain, with an area of
approximately 100,000 km$^2$, and includes three geological units, the Pyrenees in the
north, the Iberian Range in the south, and the large depression of the Ebro Valley
separating the two (Fig. 1). The Ebro Valley has an average altitude of 200 m a.s.l. and
its climate can be characterized as Mediterranean-type, with warm summers, cold
winters and continental characteristics increasing with distance inland. Certain
geographic aspects determine its climatic characteristics; for example, several mountain
chains isolate the valley from moist winds, preventing precipitation. Thus, in the central
areas of the valley, annual precipitation is low, with small monthly variations and an
annual precipitation in the central Ebro Valley of approximately 322 mm (Serrano-
Notivoli et al., 2017). In both the Pyrenees and the Iberian Range, the main climatic
characteristics are related to a transition from oceanic/continental to Mediterranean
conditions in the east. In addition, the barrier effect of the most frequent humid air
masses causes gradually higher aridity towards the east and south (Vicente-Serrano,
2005; López-Moreno & Vicente-Serrano, 2007). Areas above 2000 m a.s.l. receive
approximately 2,000 mm of precipitation annually, increasing to 2,500 mm in the
highest peaks of the mountain range (García-Ruiz, et al., 2001). Annual precipitation in
the Mediterranean coast is higher than that in the central Ebro Valley and ranges from
approximately 500 mm in Tortosa to 720 mm in Girona (Serrano-Notivoli et al., 2017).

## 2.2. From historical documents to climate: Development of a drought index for each location in NE Spain from 1650 to 1899 AD

Historical documents from 13 cities in the northeast of Spain were compiled into a novel dataset by using a consistent approach (Fig. 1, Tab. 1, Tab. 2). These historical documents are the rogation ceremonies reported in the 'Actas Capitulares' of the municipal archives or main cathedrals. The documents (described in Table 2) range from 461 years of continuous data in Girona, to 120 years in Lleida, with an average of 311 years of data on each station. Rogations were not only religious acts but also supported by the participation of several institutions; agricultural organizations and municipal and ecclesiastical authorities analyzed the situation and deliberated before deciding to hold a rogation ceremony (Vicente-Serrano and Cuadrat, 2007). Usually, the agricultural organizations would request rogations when they observed a decrease in rainfall, which could result in weak crop development. The municipal authorities would then recognize the predicament and discuss the advisability of holding a rogation ceremony. Whether a rogation was celebrated or not was not arbitrary, since the cost was paid from the public coffers. When the municipal authorities decided to hold a rogation, the order was communicated to the religious authorities, who placed it on the calendar of religious celebrations and organized and announced the event. Previous studies have reported that winter precipitation is key for the final crop production in dry-farming areas of the Ebro Valley (wheat and barley; Austin et al., 1998a, 1998b; McAneney and Arrué, 1993; Vicente-Serrano and Cuadrat, 2007). In addition to winter rogations, most of the others were held during the period of crop growth (March-May) and harvesting (June-August), since the socio-economic consequences when the harvest was poor were more evident at those times. Thus, it is reasonable to view rogations in an index from December to August. Finally, from the various types of droughts, we will be referring to a combination between meteorological and agricultural droughts. The rogation was not only agronomical or focused on a drought or agricultural problem. They already inferred that the problem was meteorological and therefore they always asked for timely rain, appropriate rain, or consistent rain. In other words, they asked for the occurrence of a meteorological phenomenon. In consequence, the follow-up or sentinel that gives them information is agricultural, but their answer is by a meteorological anomaly, and they ask for the development of a normalized meteorology, that in consequence will allow a development of the appropriate agriculture.

The qualitative information contained in the rogations was transformed into a semi-quantitative, continuous monthly series following the methodology of the Millennium Project (European Commission, IP 017008-Domínguez-Castro et al., 2012). Only *pro-pluviam* rogations were included in this study. According to the intensity of the religious act, which were uniform ceremonies performed throughout the Catholic territories and triggered by droughts, we categorized the events in 4 levels from low to high intensity: 0, there is no evidence of any kind of ceremony; 1, a simple petition within the church was held; 2, intercessors were exposed within the church; and 3, a procession or pilgrimage took place in the public itineraries, the most extreme type of rogation (see Tab. 3). Although rogations have appeared in historical documents since the late 15[th]

century and were reported up to the mid-20<sup>th</sup> century, we restricted the common period to 1650-1899 AD, since there are a substantial number of data gaps before and after this period, although some stations do not cover the full period. A continuous drought index (DI) was developed for each site by grouping the rogations at various levels. A simple approach, similar to that of Martín-Vide and Barriendos (1995) and Vicente-Serrano and Cuadrat (2007), was chosen. The annual DI values were obtained by determining the weighted average of the number of levels 1, 2 and 3 rogations recorded between December and August in each city. The weights of levels 1, 2 and 3 were 1, 2, and 3, respectively. Accordingly, the drought index for each city is a continuous semi-quantitative value from 0, indicating the absence of drought, to a maximum of 3 (Figure 2A).

## 2.3.    Clustering station drought to regional drought indices from 1650 to 1899 AD

To evaluate similarities among local stations, we performed a cluster analysis (CA) that separates data into groups (clusters) with minimum variability within each cluster and maximum variability between clusters. We selected the period of common data 1650-1770 to perform the cluster analysis. The main benefit of a cluster analysis (CA) is that it allows similar data to be grouped together, which helps to identify common patterns between data elements. To assess the uncertainty in hierarchical cluster analysis, the R package 'pvclust' (Suzuki and Shimodaira, 2006) was used. We used the Ward's method in which the proximity between two clusters is the magnitude by which the summed squares in their joint cluster will be greater than the combined summed square in these two clusters SS12−(SS1+SS2) (Ward, 1963; Everitt et al., 2001). Next, the root of the square difference between co-ordinates of a pair of objects was computed with its Euclidian distance. Finally, for each cluster within the hierarchical clustering, quantities called p-values were calculated via multiscale bootstrap resampling (1000 times). Bootstrapping techniques do not require assumptions such as normality in original data (Efron, 1979) and thus represent a suitable approach to the semi-quantitative characteristics of drought indices (DI) derived from historical documents. The *p-value* of a cluster is between 0 and 1, which indicates how strongly the cluster is supported by the data. The package 'pvclust' provides two types of *p-values*: AU (approximately unbiased *p-value*) and BP (bootstrap probability) *value*. AU *p-value* is computed by multiscale bootstrap resampling and is a better approximation of an unbiased *p-value* than the BP value computed by normal bootstrap resampling. The frequency of the sites falling into their original cluster is counted at different scales, and then the *p-values* are obtained by analyzing the frequency trends. Clusters with high AU values, such as those >0.95, are strongly supported by the data (Suzuki and Shimodaira, 2006). Therefore, in this study, sites belonging to the same group were merged by means of an arithmetical average (Eq.1).

Eq.1 $Regional\ Drought\ Index\ (\bar{x}) = (x_1 + x_2 + x_3 \ldots)/\text{n}$

where $x_n$ represents each individual annual drought index, and n is the number of
drought indices per cluster. To evaluate the relationship of each site's rogations, we then
performed a matrix correlation (Spearman) between the new groups derived from the
cluster and each individual drought index for the 1650-1899 period.

## 2.4. Validation of the regional drought indices against overlapping instrumental series.

To better understand the relationship between the derived drought indices and the
instrumental series, we used the longest instrumental precipitation and temperature
series covering the period 1786-2014 AD (Prohom et al., 2012; Prohom et al., 2015) for
the city of Barcelona and thus overlapping the rogation ceremony period of the local DI
of Barcelona (DIBARCELONA) from 1786 to 1899 AD. However, the instrumental series
was homogenized and completed including data from cities nearby and along the
Mediterranean coast (see Prohom et al., 2015 for details). Therefore, the instrumental
series contains coherent regional information from a Mediterranean section similar to
our regional drought indices stations located along the Mediterranean coast. We then
calculated the Standardized Precipitation Index (SPI, McKee et a., 1993) and the
Standardized Evapotranspiration and Precipitation Index (SPEI, Vicente-Serrano et al.,
2010). SPEI was calculated with the R Package 'SPEI' (Begueria et al., 2014). From the
various ways of calculating evapotranspiration we chose Thornwaite, which only
requires temperature and latitude as input. Next, we calculated the Spearman
correlation between the drought indices of the Mediterranean coast and the SPI/SPEI at
different time scales including a maximum lag of 12 months covering the period 1787-
1899. Further exploration of the relationship between the drought indices inferred from
historical documents and the instrumental drought indices through time were
performed by 30- and 50-year moving correlations. Finally, to avoid the circularity
problem we performed the same analysis leaving one local station out each time.

## 2.5. Detecting extreme drought years and periods in the north-east of Spain between 1650-1899 AD and links to large-scale volcanic forcing

To identify the extreme drought years, we selected those above the 99[th] percentile
of each regional drought index and mapped them in order to find common spatial
patterns. In addition, the 11-year running mean performed for each drought index
helped highlight drought periods within and among the drought indices. Finally, since
rogation ceremonies are a response of the population to an extreme event, we
performed a superposed epoch analysis (SEA; Panofsky and Brier, 1958) of the three
years before and after the volcanic event, using the package 'dplR' (Bunn, 2008) to
identify possible effects on the hydroclimatic cycle caused by volcanic eruptions. The
method involves sorting data into categories dependent on a key-date (volcanic events).
For each category, the year of the eruption is assigned as year 0, and we selected the
values of the drought indices for the three years prior to the eruption and three years
following in order to obtain a SEA matrix (number of volcanic events multiplied by 7).
For each particular event, the anomalies with respect to the pre-eruption average were
calculated to obtain a composite with all the events for the 7 years. Statistical

significance of the SEA was tested by a Monte-Carlo simulation based on the null hypothesis of finding no association between the eruptions and the climatic variables studied. Random years are chosen for each category as pseudo-event years, and the average values are calculated for -3 to +3, the same as for real eruptions. This process is repeated to create 10,000 randomly-generated composite matrices, which are sorted, and a random composite distribution is created for each column in the matrix (i.e. year relative to the eruption year 0). The distributions are then used to statistically compare the extent to which the existing composites are anomalous. We used these distributions to test the significance of the actual composites at a 99% confidence level. The largest volcanic eruptive episodes (Sigl et al., 2015) chosen for the analysis were 1815, 1783, 1809, 1695, 1836, 1832, 1884 and 1862. In addition, we performed the SEA only with the largest eruption of this period, the Tambora eruption in the year 1815.

## 3. Results

### 3.1. From historical documents to climate: Development of a drought index for each location in NE Spain from 1650 to 1899 AD

We converted the ordinal data into continuous semi-quantitative index data by performing a weighted average of the monthly data (see methods). As a result, we developed an annual drought index (from the previous December to the current August) containing continuous values from 0 to 3 collected from information on the annual mean extreme droughts of each year for each of the 13 locations. The empirical cumulative distribution function (EDCF, Fig.2A) confirmed that the new drought indices can be treated as a continuous variable, since the drought index can take almost infinite values in the range from 0 to 3 (Fig.2B). To study drought across the region, we performed a cluster analysis including the annual drought indices of the 13 cities. These data were then used to study the hydrological responses after strong tropical eruptions.

### 3.2. Clustering station drought to regional drought indices from 1650 to 1899 AD

The cluster analysis (CA, see methods) using the DI of the 13 locations and after applied to the complete period until 1899 revealed three significant and physically coherent areas, hereafter known as Mountain, Mediterranean and Ebro Valley (Fig. 3). The first cluster includes cities with a similar altitude (Teruel, La Seu) and similar in latitude (Barbastro, Lleida, Huesca, Girona, see Fig. 1). The cities within the second and third clusters are near the Ebro River (Calahorra, Zaragoza and Tortosa) or have similar climatic conditions (Cervera, Vic, Barcelona, Tarragona). Clusters two and three suggest (Fig. 3) that the coherence of the grouping can be explained by the influence and proximity of the Mediterranean Sea (Tortosa, Cervera, Tarragona, Vic and Barcelona) and the influence of a more continental climate (Zaragoza and Calahorra). Accordingly, three regional drought indices were developed by combining the individual DIs of each group; DI Mountain (DIMOU), composed of Barbastro, Teruel, Lleida, La Seu, and Girona; DI Mediterranean (DIMED), composed of Tortosa, Cervera, Tarragona, Vic and

Barcelona, and DI Ebro Valley (DIEV), comprising Zaragoza and Calahorra. The resulting
drought indices in regional DI series can also vary from 0 to 3 but show a relatively
continuous distribution range (Figure 2B).
The Spearman correlation matrix for the period 1650-1899 AD confirms the high
and significant (p<0.05) correlations between each individual DI and its corresponding
group, confirming the validity of the new DI groups (Fig. 4). The correlations among the
cluster drought indices range from 0.76 (between DIEV and DIMED) to r=0.38 (between
DIEV and DIMOU) and r=0.42 (between DIMED and DIMOU). In DIEV, both of the local
DIs show similar correlations (Zaragoza, r=0.73; Calahorra, r=0.75). In the DIMED cluster,
the high correlations among the members show strong coherency. DIMOU is the most
heterogeneous cluster, with correlations of r=0.57 for Barbastro and r=0.33 for La Seu.
Although each individual DI within this group and within the DIMOU shows significant
correlation, individual DIs compared one to another reveal some correlation values not
to be significant (p<0.05).

**3.3.    Validation of the regional drought indices against overlapping instrumental
series.**

The highest Spearman correlation (r=-0.46; p<0.001) between the Barcelona
drought index and the instrumental SPI over the full 113-year period (1787-1899 AD;
Fig.5C) was found for the SPI of May with a lag of 4 months ($SPI_{MAY\_4}$ hereafter). A slightly
lower, though still significant correlation was obtained from the SPEI of May with a lag
of 4 months ($SPEI_{MAY\_4}$) (r=-0.41; p<0.001, Fig.5D). The regional Mediterranean drought
index shows moderately higher correlations with the instrumental SPI (r=-0.53; p<0.001)
and SPEI (r=-0.50; p<0.001) computed for the same period and time scale. The moving
correlations analyses between DIMED, DIBARCELONA and $SPI_{MAY\_4}$ for 30 and 50 years
(Fig.5A; Fig.5B) presented significant values through the full period. However, the
agreement is especially higher and stable during the period 1787-1834. After 1835
despite that correlations remain significant, the instability is higher, and the agreement
decreased.
Furthermore, when the analysis was performed leaving one station out each time
(Fig. S1), the results remain significant (p<0.001) and the correlation in all cases is above
0.45. The next step (iv) will address the selection of extreme drought years and periods
within the 250 years from 1650-1899 AD using information from the cluster analysis.

**3.4.    Detecting extreme drought years and periods in the north-east of Spain
between 1650-1899 AD and links to large-scale volcanic forcing**

According to the cluster grouping, the three new spatially averaged drought
indices (DIEV, DIMED and DIMOU) are presented in Fig. 6. Mountain DI (DIMOU) had the
least number of drought events and a maximum DI of 1.6 in 1650 AD. The Ebro Valley DI
(DIEV) had the highest number of droughts (derived from the highest number of positive
index values) followed by the third region (Mediterranean DI, DIMED). The 17[th] and 18[th]
centuries exhibited a relatively large number of severe droughts (Fig. 6). High positive
index values over the duration of the DIs in all three series indicate that a drought period
occurred from 1740 to 1755 AD. The lowest DIs were found at the end of the 19th
century, meaning that droughts were less frequent in this period. The 11-year running
mean shows common periods with low DI values, such as 1706-1717, 1800-1811, 1835-
1846 and 1881-1892, which we infer to be 'normal' or drought-free. On the other hand,
1678-1689, 1745-1756, 1770-1781, and 1814-1825 are periods with continuously high
DIs, indicating that significant droughts affected the crops during these periods and
intense rogation ceremonies were needed.

In the Ebro Valley, the most extreme years (Fig. 6) (according to the 99%
percentile of the years 1650-1899) were 1775 (drought index value of 2.8), 1798 (2.7),
1691 (2.6), 1753 (2.5) and 1817 (2.5). Most of these extreme drought years can also be
found in DIMED 1753 (2.6), 1775 (2.5), 1737 (2.3), 1798 (2.2) and 1817 (2.2). In DIMOU,
the extreme drought years occurred in the 17th century: 1650 (1.6), 1680 (1.5), 1701
(1.5) and 1685 (1.4), and are spatially displayed in Fig. 7. In the years 1775 and 1798, the
Ebro Valley, Mediterranean and some mountain cites suffered from severe droughts. It
is notable that the year 1650 in the Mountain area presented high values of DI, while
the other locations had very low DI values (DIEV=0.4; DIMED=0.8).

We performed a superposed epoch analysis (SEA, see methods) to study the
drought response over north-east Iberia to major volcanic eruptions (Fig. 8a). The figure
shows significant decreases ($p<0.05$) in the Ebro Valley and Mediterranean DI values
during the year a volcanic event occurred and for the following year. We did not find a
post-volcanic drought response in the Mountain area. No significant response was found
for any of the DIs two or three years after the volcanic eruptions, including the major
ones. However, two years after the Tambora eruption in April 1815, there was a
significant ($p<0.05$) increase in the three drought indices (DIEV, DIMED and DIMOU) (Fig.
8b).

## 4. Discussion

In the northeast Iberian Peninsula, drought recurrence, intensity, persistence
and spatial variability have mainly been studied by using instrumental data covering the
past ca. 60 years (Vicente-Serrano et al., 2014; Serrano-Notivoli et al., 2017). In addition,
natural proxy data, including specially tree-ring chronologies, have been used to infer
drought variability before the instrumental period (Esper et al., 2015; Tejedor et al.,
2016, 2017c; Andreu-Hayles et al., 2017). Nevertheless, most of such highly temporally
resolved natural proxy-based reconstructions represent high-elevation conditions
during specific periods of the year and as a consequence, drought behavior in large low
elevation areas remains poorly explored. In these areas however, documentary records
as rogation ceremonies, have demonstrated potential to complement the
understanding of droughts across Europe (e.g. Brázdil et al., 2005, 2010, 2018).

Still, rogation ceremonies need to be considered as a "cultural" proxy affected by
a certain degree of subjectivity due to the perception of people about hydroclimate
events. In consequence, the analysis must be cautious, taking into account their
historical and sociological nature. Further limitations are related to their binomial
character (occurrence or not of rogation ceremonies), the cumulative character of
drought and then the difficulty of the interpretation of sequential rogations or the
restrictions to perform a rigorous calibration-verification approach due to a lack of
overlapping periods with observational weather series.
Despite these limitation, and potential variations in the timing of occurrence of
rogations in different areas or periods due to differences/variations in agricultural
practices, we developed drought indices (DI) derived from rogations occurred from early
winter to August that can be considered as reliable drought proxies (even if only in some
environments and some specific historical periods). More specifically, we found that i)
DI series exhibit a coherent regional pattern but their reliability is lower in mountain
areas, ii) Represent a useful climate proxy for at least the period 1650-1830`s but its
reliability decreases thereafter.
Due to the cumulative character of drought, the delays between drought and
rogation occurrence and their differential influence on different agricultural species and
environmental conditions an accurate definition of the temporal scale in drought that is
represented by the rogation is challenging. In this paper, for comparative purposes, a
conservative approach is used by combining rogations occurred from December to
August in an index trying to account for general drought conditions occurred during the
whole crop growing season across the whole study area (spring and summer) but also
including previous conditions that may have impact in final production (spring and
winter rogations are likely to reflect drought conditions occurred in winter and previous
autumn).
Further limitations when dealing with historical documents as a climatic proxy
are related to converting binomial qualitative information (occurrence or not of rogation
ceremonies) into quantitative data (e.g. Vicente-Serrano and Cuadrat, 2007;
Dominguez-Castro et al., 2008). Here, we followed the methodology proposed in the
Millennium Project (European Commission, IP 017008) and also applied in Domínguez-
Castro et al., (2012). According to such proceedings and considering both the occurrence
or otherwise of rogation ceremonies and the intensity of the religious acts, the
information contained in historical documents can be transformed into a semi-
quantitative time series (including continuous values from 0 to 3). To that extent, the
ECDF analysis helped in understanding the nature of the historical documents when
transformed into semi-quantitative data, confirming that they can be treated as a
continuous variable. We then aggregated the annual values to develop a continuous
semi-quantitative drought index (DI) where values can range from zero (absence of
drought) to a maximum of 3 (severe drought). This set of procedures technically solves
the structural problem of the data. However, we have added complexity to its
interpretation since, for example, an index of level 2 does not necessarily imply that a
drought was twice as intense as a drought classified as level 1, nor that the change in
the intensity of droughts from level 1 to level 2 or from level 2 to 3 has to be necessarily
equivalent. Yet, we can infer with much confidence that if there was a drought of level
2 it is because those types of ceremonies of level 1, if occur, did not work, and therefore
the drought was still an issue for the development of the crops i.e., there is a progressive
drying, but it does not have to be twice as intense. Hence, this must be taken into
account when interpreting the indices.

The confirmation of rogation ceremonies as a valid drought proxy requires an
additional procedure -the calibration/verification approach. However, continuous
rogation documents end in the 19[th] century, whereas instrumental weather data
generally begins in the 20[th] century (Gonzalez-Hidalgo et al., 2011). In the study area,
only the continuous and homogenized instrumental temperature and precipitation
series of Barcelona (Prohom et al., 2012; 2015) overlap the existing drought indices. Our
results suggest that rogation ceremonies are not only valid as local indicators (good
calibration/ verification with the local DIBARCELONA), but they also have regional
representativeness (DIMED) and provide valuable climatic information (good
calibration/ verification with the regional DIMED). To the best of our knowledge, this is
the first time that rogation ceremonies in the Iberian Peninsula have been calibrated
with such a long instrumental period. The correlation is maximized in May, the key
month for the harvest to develop properly. In addition, the 4-month lag confirms the
importance of the end of winter and spring precipitation for good crop growth. The high
DIMED correlation (r=-0.53; $p$<0.001) indicates not only that this cluster captures the
Mediterranean drought signal, but also that it can be used as a semi-quantitative proxy,
with verification results similar to the standards required in dendroclimatology (Fritts et
al., 1990).

In spite of being statistically valid for the whole analyzed period, the suitability
of the drought index significantly varies in time. The agreement with instrumental
weather data is especially higher during the period 1787-1834 but decrease thereafter.
It is challenging to determine whether the decrease in the number of rogations after
1835 is due to the lack of droughts, the loss of documents, or a loss of religiosity. For
instance, after the Napoleonic invasion (1808-1814) and the arrival of new liberal
ideologies (Liberal Triennial 1820-1823), there was a change in the mentality of people
in the big cities. These new liberal ideas were concentrated in the places where
commerce and industry began to replace agriculturally based economies, leading to
strikes and social demonstrations demanding better labor rights. New societies were
less dependent on agriculture; hence, in dry spells, the fear of losing crops was less
evident and fewer rogations were performed. In short, the apparent decrease of
rogations in the 19[th] century could be explained by a combination of political instability
in the main cities and the loss of religiosity and historical documents. Nevertheless, the
institutional controls in pre-industrial society were so strict that many of its constituent
parts remained unchanged for centuries, and rogation ceremonies are one of such
elements. This can be explained by two different factors. First, rogation ceremonies are
used within the framework of the Roman Church Liturgy, so changes can only be defined
and ordered by the Vatican authorities. If there is a will to change criteria affecting the
substance of liturgical ceremonies, all involved institutions must record considerations,
petitions and decisions in official documents from official meetings, supported by public
notaries. In addition, changes must be motivated from the highest institutional level
(Pope) to the regional authorities (Bishops) and local institutions (Chapters, parishes...).
This system was too complex to favor changes. A second mechanism guarantees the
stability of the rogation system: if any minor or important change in rogations was
instigated at local level by the population or local institutions, this interference directly
affected the Roman Church Liturgy. Then, it was a change not to be taken lightly as the
Inquisition Court would start judicial proceedings and could bring a criminal charge of
heresy. The punishment was so hard that neither institutions nor the people were
interested in introducing changes in rogations.
To further calibrate the potential of this source of information as a climatic proxy,
we need to consider the existence of coherent spatial patterns in the distribution of
droughts. The instrumental climate data is subject to quality controls to determine the
extent to which patterns reflect elements of the climatic cycle or may be due to errors
of measurement, transcription of information etc (e.g. Alexanderson, 1986). In this
paper, the local series are compared with the regional reference series as a basic
element of quality control (e.g. Serrano-Notivoli et al., 2017). The interpretation of other
proxies, such as tree-ring records are subject to similar quality control procedures to
guarantee the spatial representativeness of the information they contain (e.g. Esper et
al., 2015; Duchesne et al., 2017; Tejedor et al., 2017c).
We were aware of the potential drawbacks and dealt with the problem of analyzing
the spatial representativeness of the rogation series through a cluster analysis. We thus
identified the extent to which the local rogation series show similar patterns to those
observed in neighboring records and can, therefore, be considered as representative of
the climate behavior at a sub-regional scale. Clustering is a descriptive technique (Soni,
2012), the solution is not unique, and the results strongly rely upon the analyst's choice
of parameter. However, we found three significant ($p<0.05$) and consistent structures
across the drought indices based on historical documents. DIEV shows a robust and
coherent cluster associated with droughts in the Ebro Valley area, including the cities of
Zaragoza and Calahorra. The high correlation among the local drought indices suggests
an underlying coherent climatic signal. DIMED shows also a robust and coherent cluster
associated with droughts in the Mediterranean coast area, including high correlation
between the local drought Indices of Tortosa, Tarragona, Barcelona, Vic and Cervera.
The high correlation between DIEV and DIMED suggests similar climatic characteristics.
Furthermore, the main cities among these two clusters share similar agrarian and
political structures that support the comparison. Still, we know from observations that,
although DIEV and DIMED locations have similar climatic characteristics, the
Mediterranean coast locations have slightly higher precipitation totals, which is
supported by the cluster. One is reflecting the Ebro Valley conditions and the other is
reflecting a more Mediterranean-like climate. Therefore, our final grouping is not only
statistically significant, but it has also a geographical/physical meaning.
We found that DIMON shows a less robust and complex structure. This cluster
includes local drought indices located in mountain or near mountain environments.
Although there is a high correlation between the local DIs and the regional DIMOU
suggesting a common climatic signal, the low correlation among local drought indices
might be explained by the fact that the productive system of the mountain areas is not
only based on agriculture, but also on animal husbandry, giving them an additional
resource for survival in cases of extreme drought. Therefore, the DIMOU cluster might
not only be collecting climatic information but also diverse agricultural practices or even
species, translated into a weaker regional common pattern. For instance, Cervera and
Lleida share similar annual precipitation totals, but belong to the Mediterranean and the
Mountain drought indices respectively. Lleida is located in a valley with an artificial
irrigation system since the Muslim period, which is fed by the river Segre (one of the
largest tributaries to the Ebro river). The drought in the Pyrenees is connected with a
shortage of water for the production of energy in the mills, as well as to satisfy irrigated
agriculture. However, the irrigation system itself allowed Lleida to manage the resource
and hold out much longer. Therefore, only the most severe droughts, and even those in
an attenuated form, were perceived in the city. Cervera, located in the Mediterranean
mountains, in the so-called pre-littoral system and its foothills, has a different
precipitation dynamic that is more sensitive to the arrival of humid air from the
Mediterranean. In addition, Lleida had a robust irrigation system that Cervera did not
have. The droughts in Cervera are more akin to the "Mediterranean" ones and thus its
presence in the Mediterranean drought index seems to be consistent.
DIMOU has a weaker climatological support and thus it should be interpreted with
particular caution. Yet, this important constraint in the interpretation of DIMOU is not
problematic from a practical point of view, since it represents an area in which there are
other proxy records (e.g. tree-rings) covering a wide spatio-temporal scale and valuable
as drought proxies (e.g. Tejedor et al., 2016; 2017c). The consistency of the clusters in
the Ebro Valley and the coastal zones (DIMED and DIEV) is especially encouraging and
reflects the high potential of rogations as a drought proxy. It is precisely in these areas
that there are no relict forests, due to human intervention, and therefore no centennial
tree-ring reconstructions can be performed to infer past climates. Consequently, in
these environments, the information from historical documents is especially relevant.
These findings open a new line of research that the authors will continue exploring
in future studies. We believe that these results highlight the validity of the drought
indices to be taken as continuous variables. In addition, the analysis confirmed that the
grouping made by the cluster analysis demonstrates spatial coherency among the
historical documents. For some places such as the mountain areas, where the
population had other ways of life in addition to agriculture, *pro-pluviam* rogation
ceremonies may have a weaker climatic significance. However, *pro-pluviam* rogations
may be especially relevant in valleys and coastal areas where there are no other climatic
proxies. The exploration of historical documents from the main Cathedrals or municipal
city archives, the Actas Capitulares, yielded the different types and payments of the
rogation ceremonies that were performed in drought-stressed situations.
Despite general limitations, our results are comparable and in agreement with
other drought studies based on documentary sources describing the persistent drought
phase affecting the Mediterranean and the Ebro Valley areas in the second half of the
18th century (as found in Vicente-Serrano and Cuadrat, (2007) for Zaragoza). The results
for the second half of the 18th century also agree with the drought patterns previously
described for Catalonia (Barriendos, 1997, 1998; Martín-Vide and Barriendos, 1995).
Common drought periods were also found in 1650-1775 for Andalusia (Rodrigo et al.,
1999, 2000) and in 1725-1800 for Zamora (Domínguez-Castro et al., 2008). In general,
based on documentary sources from Mediterranean countries, the second half of the
18th century has the highest drought persistency and intensity, which may be because
there were more blocking situations in this period (Luterbacher et al. 2002, Vicente-
Serrano and Cuadrat, 2007). The period of 1740-1800 AD coincides with the so-called
'Maldá anomaly period'; a phase characterized by strong climatic variability, including
extreme drought and wet years (Barriendos and Llasat, 2003). The 18th century is the
most coherent period, including a succession of dry periods (1740-1755), extreme years
(1753, 1775 and 1798) and years with very low DIs, which we interpret as normal years.
Next, the period from 1814-1825 is noteworthy due to its prolonged drought. The causes
of this extreme phase are still unknown although Prohom et al. (2016) suggested that
there was a persistent situation of atmospheric blocking and high-pressure conditions
at the time.
Results are also in line with described hydroclimatic responses to volcanic
forcing. In the Ebro Valley and the Mediterranean area, rogation ceremonies were
significantly less frequent in the year of volcanic eruptions and for the following year.
Such patterns may be explained by the volcanic winter conditions, which are associated
with reductions in temperature over the Iberian Peninsula 1-3 years after the eruption
(Fischer et al., 2007; Raible et al., 2016). The lower temperature is experienced in spring
and summer after volcanic eruptions compared to spring and summer conditions of non-
volcanic years. This might be related to a reduction in evapotranspiration, which reduces
the risk of droughts. This reinforces the significance of volcanic events in large-scale
climate changes. Furthermore, a significant increase in the intensity of the droughts was
observed two years after the Tambora eruption in the three clusters (Fig.8) in agreement
with findings by Trigo et al., (2009). This result is similar to that of a previous study using
rogation ceremonies in the Iberian Peninsula, although it was based on individual and
not regional drought indices (Dominguez-Castro et al., 2010). In addition, the normal
conditions in the year of the Tambora eruption and the following year, and the increased
drought intensity two years after the event, are in agreement with recent findings on
hydroclimatic responses after volcanic eruptions (Fischer et al., 2007; Wegmann et al.,
2014; Rao et al., 2017; Gao and Gao 2017), although based on tree ring data only. In
addition, Gao and Gao, (2017) highlight the fact that high-latitude eruptions tend to
cause drier conditions in western-central Europe two years after the eruptions. Rao et
al., (2017) suggested that the forced hydroclimatic response was linked to a negative
phase of the East Atlantic Pattern (EAP), which causes anomalous spring uplift over the
western Mediterranean. This pattern was also found in our drought index for the
Tambora eruption (1815 AD), but no significant pattern was found in north-east Spain
for the other major (according to Sigl et al., 2015) volcanic eruptions. In particular, the
mountain areas show less vulnerability to drought compared to the other regions. This
is mainly due to the fact, that mountainous regions experience less evapotranspiration,
more snow accumulation and convective conditions that lead to a higher frequency of
thunderstorms during the summertime. Volcanic forcing, however, may differentially
modulate seasonal climate conditions by their influence on the North Atlantic Oscillation
and in the East Atlantic circulation patterns. This seasonal detail cannot be clarified in
our research due to the annual scale used to compute the drought indices.

## 5. Conclusions

We developed a new dataset of historical documents by compiling historical
records (rogation ceremonies) from 13 cities in the northeast of the Iberian Peninsula.
These records were transformed into semi-quantitative continuous data to develop
drought indices (DIs). We regionalized them by creating three DIs (Ebro Valley,
Mediterranean and Mountain) covering the period from 1650 to 1899 AD. The intensity
of the DI is given by the strength and magnitude of the rogation ceremony, and the
spatial extent of the DI is given by the cities where the rogations were held.
Our study highlights three considerations: i) the spatial and temporal resolution
of rogations should be taken into account, particularly when studying specific years,
since the use of *pro-pluviam* rogations gives information about drought periods and not
about rainfall in general. Accordingly, it must be stressed that the drought indices
developed here are not precipitation reconstructions; rather, they are high-resolution
extreme event reconstructions of droughts spells. The comparison of these results with
other continuous proxy records must be carried out with caution (Dominguez-Castro et
al., 2008), although here we found a very high and stable correlation with the
instrumental series for the overlapping period, which opens new lines of research. ii)
The validity of rogation ceremonies as a high-resolution climatic proxy to understand
past drought variability in the coastal and lowland regions of the north-eastern
Mediterranean Iberian Peninsula is clearly supported by our study. This is crucial,
considering that most of the high-resolution climatic reconstructions for the northern
Iberian Peninsula have been developed using tree-ring records collected from high-
elevation sites (>1,600 m a.s.l.) in the Pyrenees (Büntgen et al., 2008, 2017; Dorado-
Liñán et al., 2012) and the Iberian Range (Esper et al., 2015, Tejedor et al., 2016, 2017a,
2017b, 2017c), to deduce the climate of mountainous areas. iii) Particularly in the
Mediterranean and in the Ebro Valley areas, significant imprints of volcanic eruptions
are found in the drought indices derived from the rogation ceremonies. These results
suggest that DI is a good proxy to identify years with extreme climate conditions in the
past at low elevation sites.
In addition, recent studies have emphasized the great precipitation (González-
Hidalgo, et al., 2011; Serrano-Notivoli et al., 2017) and temperature variabilities
(González-Hidalgo, et al., 2015) within reduced spaces, including those with a large

altitudinal gradient, such as our study area. Finally, the historical data from rogations covers a gap within the instrumental measurement record of Spain (i.e., which starts in the 20[th] century). Hence, rogation data are key to understanding the full range of past climate characteristics (in lowlands and coastal areas), in order to accurately contextualize current climate change. We encourage the use of further studies to better understand past droughts and their influence on societies and ecosystems; learning from the past can help to adapt to future scenarios, especially because climate variability is predicted to increase in the same regions where it has historically explained most of the variability in crop yields.

## Acknowledgments.
We would like to thank the support of all the custodians of the historical documents.

## Author contributions.
E.T., and J.M.C. conceived the study. J.M.C. and M.B. provided the data. E.T. and M.d.L. conducted the data analysis, and E.T. wrote the paper with suggestions of all the authors. All authors discussed the results and implications and commented on the manuscript at all stages.

## Competing interests.
The authors declare no competing interests.

## Financial support.
Supported by the project 'CGL2015-69985' and the government of Aragon (group Clima, Cambio Global y Sistemas Naturales, BOA 147 of 18-12-2002) and FEDER funds.

## Data availability.
The datasets generated during and/or analysed during the current study are available from the corresponding author on reasonable request.

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

**Figures and tables**

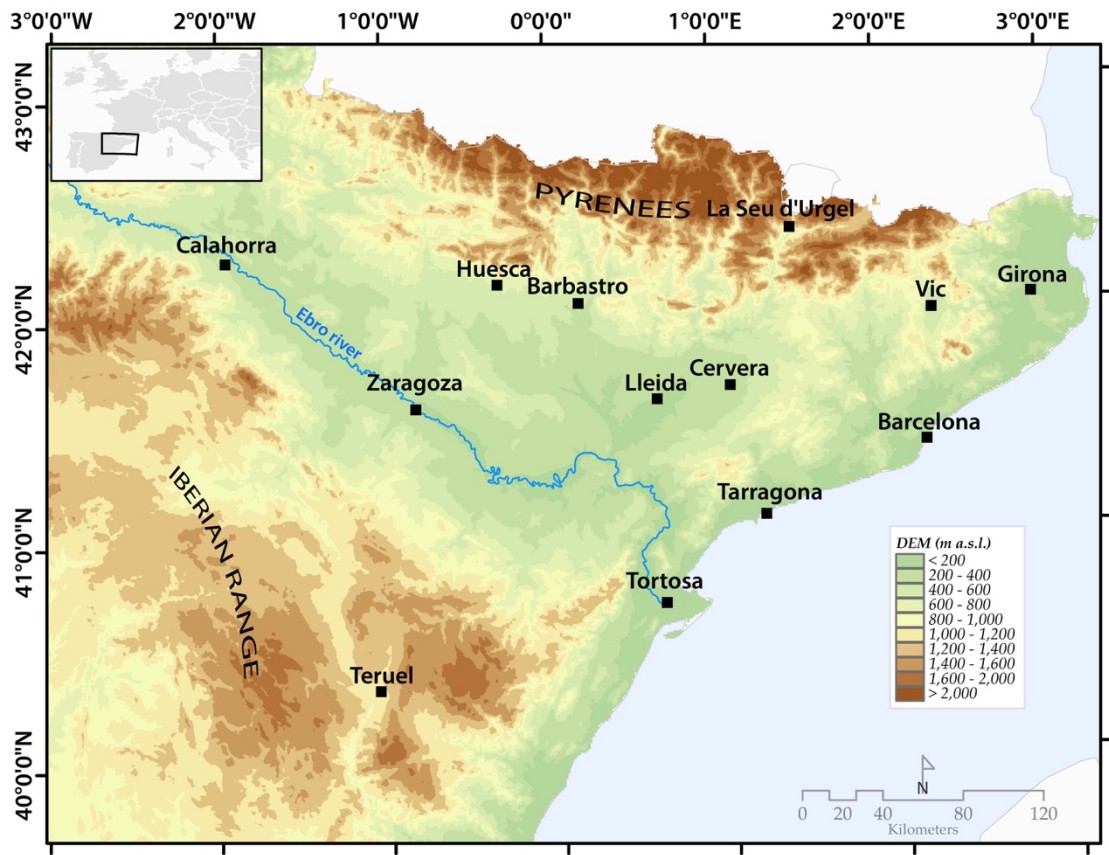


Figure 1. Location of the historical documents in the northeast of Spain.



| Site | Latitude (degrees) | Longitude (degrees) | Altitude (m.a.s.l.) | Start (Years AD) | End | Extension (years) |
|---|---|---|---|---|---|---|
| Zaragoza | 41.64 | -0.89 | 220 | 1589 | 1945 | 356 |
| Teruel | 40.34 | -1.1 | 915 | 1609 | 1925 | 316 |
| Barbastro | 42.03 | 0.12 | 328 | 1646 | 1925 | 279 |
| Calahorra | 42.3 | -1.96 | 350 | 1624 | 1900 | 276 |
| Huesca | 42.13 | -0.4 | 457 | 1557 | 1860 | 303 |
| Girona | 42.04 | 2.93 | 76 | 1438 | 1899 | 461 |
| Barcelona | 41.38 | 2.17 | 9 | 1521 | 1899 | 378 |
| Tarragona | 41.11 | 1.24 | 31 | 1650 | 1874 | 224 |
| Tortosa | 40.81 | 0.52 | 14 | 1565 | 1899 | 334 |
| LaSeu | 42.35 | 1.45 | 695 | 1539 | 1850 | 311 |
| Vic | 41.92 | 2.25 | 487 | 1570 | 1899 | 329 |
| Cervera | 41.67 | 1.27 | 548 | 1484 | 1850 | 366 |
| Lleida | 41.61 | 0.62 | 178 | 1650 | 1770 | 120 |

Table 1. Historical document characteristics in the northeast of Spain.

983

**Teruel**
• Chapter Acts of the Holy Church and Cathedral of Teruel, 1604-1928, 28 vols.
**Barbastro**
• Cathedral Archive of Barbastro 'Libro de Gestis', Barbastro (Huesca), 1598-1925, 23 vols.
**Barcelona**
• City Council Historical Archive of Barcelona (AHMB), "Manual de Novells Ardits" o "Dietari de l'Antic Consell Barceloní", 49 vols., 1390-1839.
• City Council Historical Archive of Barcelona (AHMB),"Acords", 146 vols., 1714-1839.
• City Council Administrative Archive of Barcelona (AACB), "Actes del Ple", 100 vols., 1840-1900.
• Chapter Acts of the Cathedral Historical Archive of Barcelona (ACCB), "Exemplaria", 6 vols., 1536-1814.
• More than 20 private and institutional dietaries.
**Calahorra**
• Chapter Acts of the Cathedral Historical Archive of Calahorra (La Rioja), 1451-1913, 35 vols.
• Archives of Convento de Santo Domingo 1782–1797. First volume. 158 pages.
**Cervera**
• Regional Historical Archive of Cervera (AHCC), Comunitat de preveres, "Consells", 12 vols., 1460-1899.
• Regional Historical Archive of Cervera (AHCC), "Llibre Verd del Racional", 1 vol., 1448-1637.
• Regional Historical Archive of Cervera (AHCC), "Llibres de Consells", 212 vols., 1500-1850.
**Gerona**
• City Council Historical Archive of Girona (AHMG), "Manuals d'Acords", 409 vols., 1421-1850.
**Huesca**
• Chapter Acts of the Cathedral Historical Archive of Huesca, 1557-1860, 15 vols.
**La Seu d'Urgell**
• City Council Historical Archive of La Seu d'Urgell (AHMSU), "Llibres de consells i resolucions", 47 vols., 1434-1936.
**Lleida**
• National Library of Madrid (BNM), Manuscript 18496, "Llibre de Notes Assenyalades de la Ciutat de Lleida", 1 vol.
• Chapter Acts of the Cathedral Historical Archive of Lleida (ACL), "Actes Capitulars", 109 vols., 1445-1923.
**Tarragona**
• City Council Historical Archive of Tarragona (AHMT), "Llibres d'Acords", 92 vols., 1800 1874.
• Departmental Historical Archive of Tarragona (AHPT), "Liber Consiliorum", 286 vols., 1358-1799.
• Regional Historical Archive of Reus (AHCR), "Actes Municipals", 10 vols., 1493-1618.
• Regional Historical Archive of Reus (AHCR), Comunitat de Preveres de Sant Pere, "Llibre de resolucions", 2 vols., 1450-1617.
**Tortosa**
• City Council Historical Archive of Tortosa (AHMTO), "Llibres de provisions i acords municipals", 119 vols., 1348-1855.
• Chapter Acts of the Cathedral Historical Archive of Tortosa (ACCTO), "Actes Capitulars", 217 vols., 1566-1853.
**Vic**
• Chapter Acts of the Cathedral Historical Archive of Vic (AEV, ACCV), "Liber porterii", 10 vols., 1392-1585.
• Chapter Acts of the Cathedral Historical Archive of Vic (AEV, ACCV), "Secretariae Liber", 30 vols., 1586-1909.
• City Council Historical Archive of Vic (AHMV), "Indice de los Acuerdos de la Ciudad de Vich des del año 1424", 2 vols., 1424-1833.
• City Council Historical Archive of Vic (AHMV), "Llibre d'Acords", 49 vols., 1424-1837.
**Zaragoza**
• Chapter Acts of the Cathedral Historical Archive 'Libro de Actas del Archivo de la Basílica del Pilar', 1516–1668, 17 vols. 2.600 pages.
• City Council Historical Archive of Zaragoza, 1439–1999. 1308 vols. 35.000 pages.
• City Council Historical Archive of Zaragoza. 'Libro de Actas del Archivo Metropolitano de La Seo de Zaragoza', 1475–1945. 81 vols. 12.150 pages.

Table 2. Documentary references for administrative public documentary sources used for rogation monthly indices (all documents are generated and initialed by public notaries). Noted that only the official documents are shown. Each documentary record is given reliability load with the public notary rubric that acts like secretary. This procedure is currently still in force for the same type of document, which is still generated at present time.

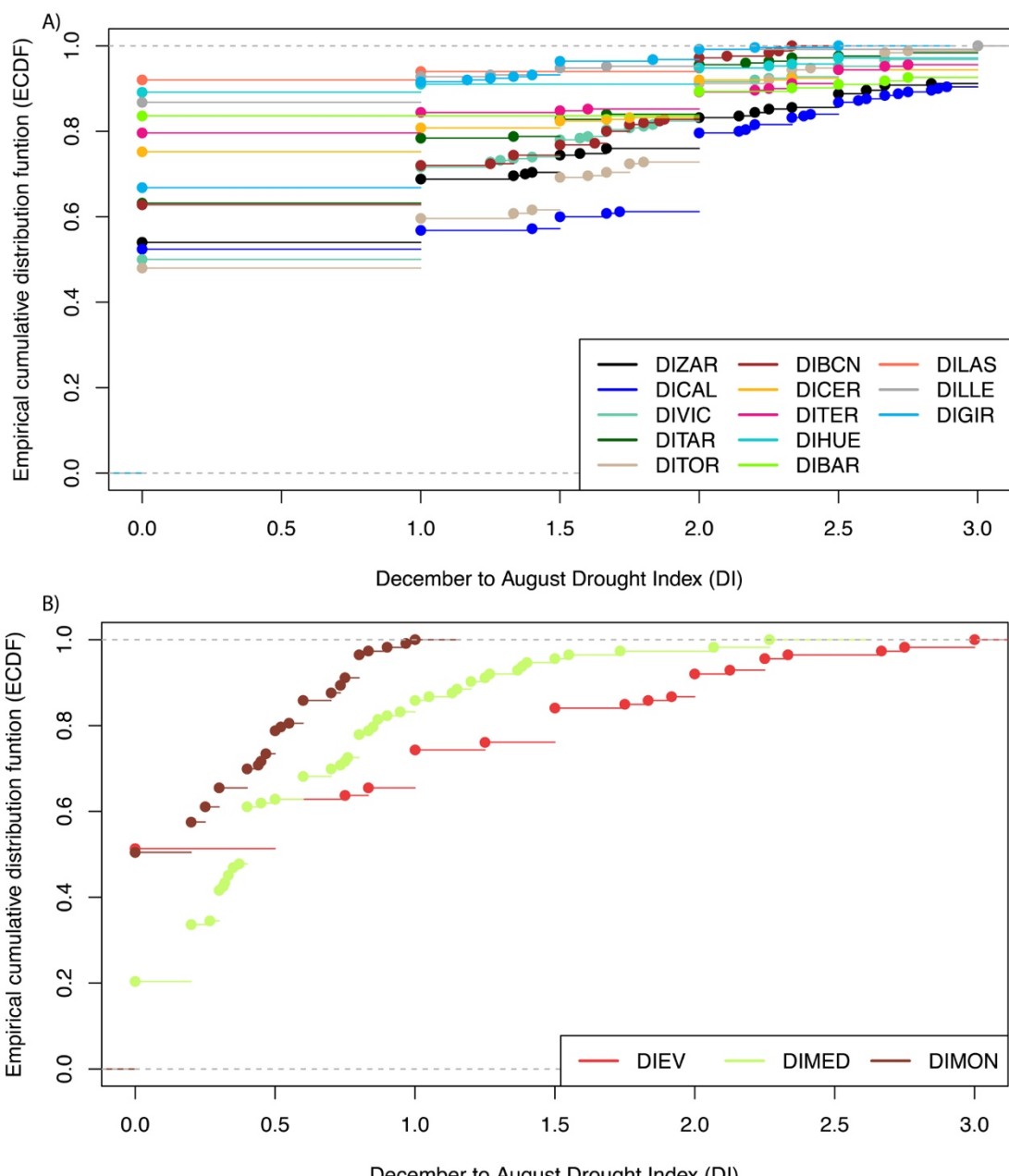


Figure 2. The empirical cumulative distribution function (ECDF), used to describe a
sample of observations of a given variable. Its value at a given point is equal to the
proportion of observations from the sample that are less than or equal to that point.
ECDF performed for the local drought indices (A) and the regional drought indices (B).
















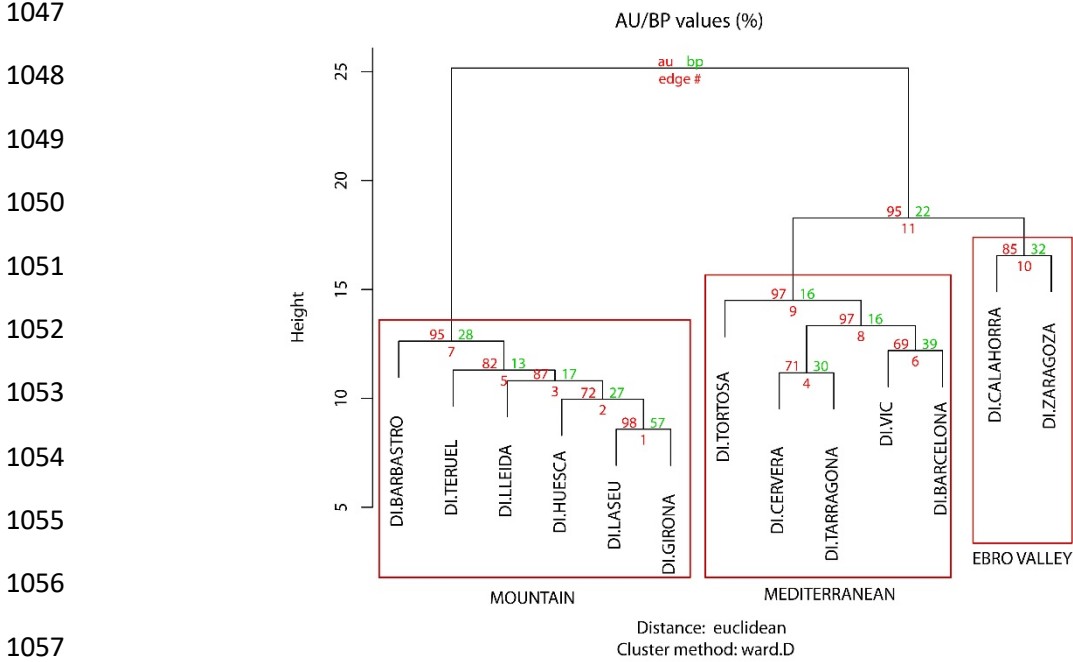

Figure 3. Dendrogram showing the hierarchical cluster analysis of the drought indices
developed from the historical documents for each location. The AU (approximately
unbiased *p-value*) is indicated in red and the BP (bootstrap probability) is presented in
green.


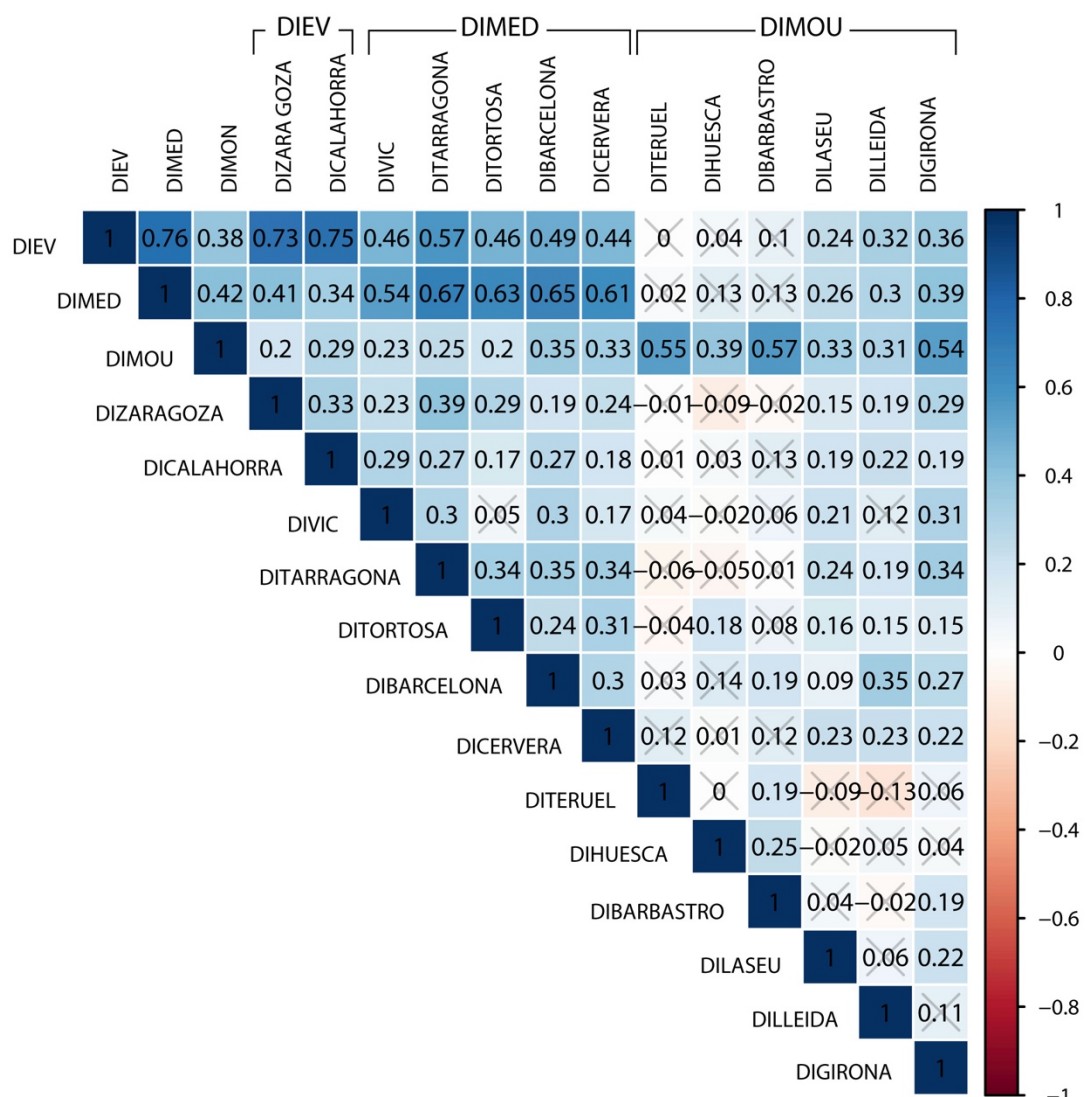


Figure 4. Correlation matrix (Spearman) between the individual drought indices and the cluster drought indices for the period of 1650-1899. Values are significant at $p<0.05$, except those marked with a gray cross, which are not significant.









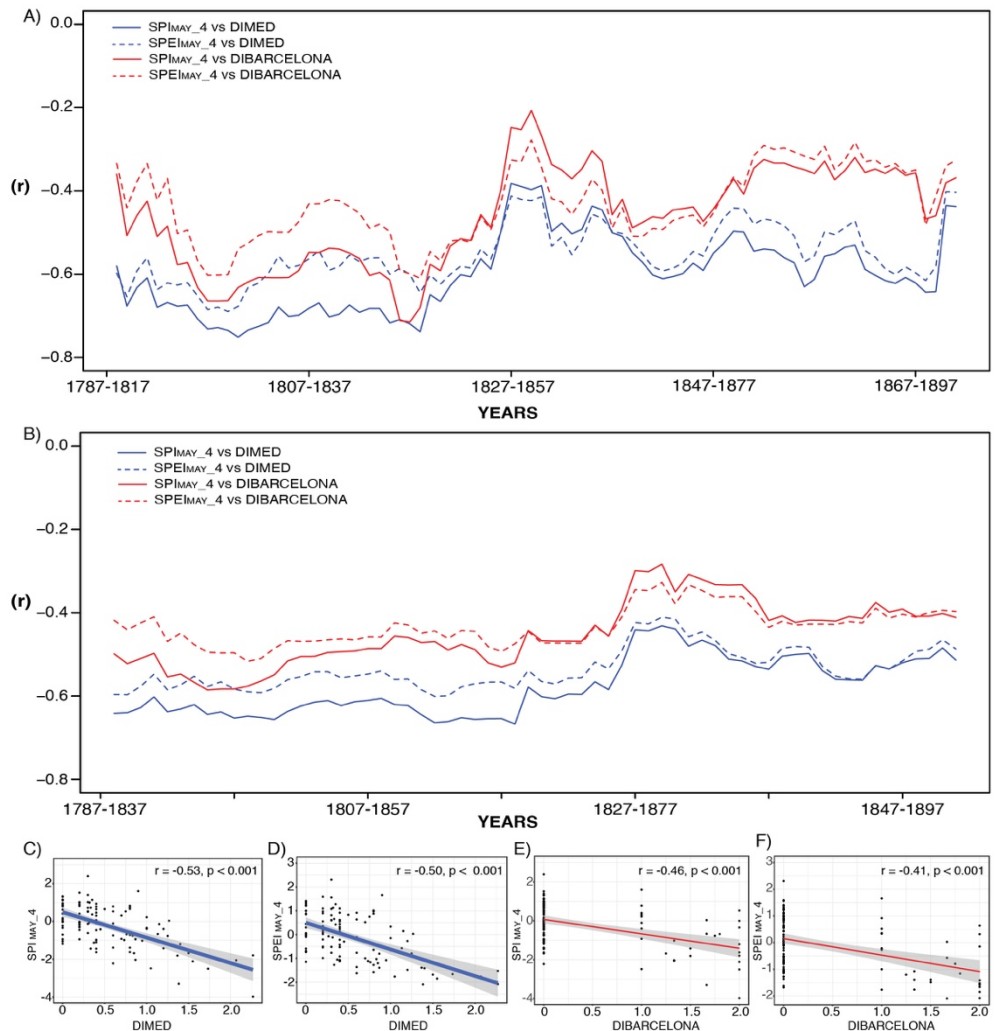


Figure 5. A) 30y moving correlation between DIMED, DIBARCELONA and the
instrumental computed SPI and SPEI. B) Same but 50y moving correlations. C)
Correlation (Spearman) between DIMED and SPI$_{MAY}$_4 for the full period (1787-1899).
D) Correlation between DIMED and SPEI$_{MAY}$_4 for the full period (1787-1899). E)
Correlation between DIBARCELONA and SPI$_{MAY}$_4 for the full period (1787-1899). F)
Correlation between DIBARCELONA and SPEI$_{MAY}$_4 for the full period (1787-1899).



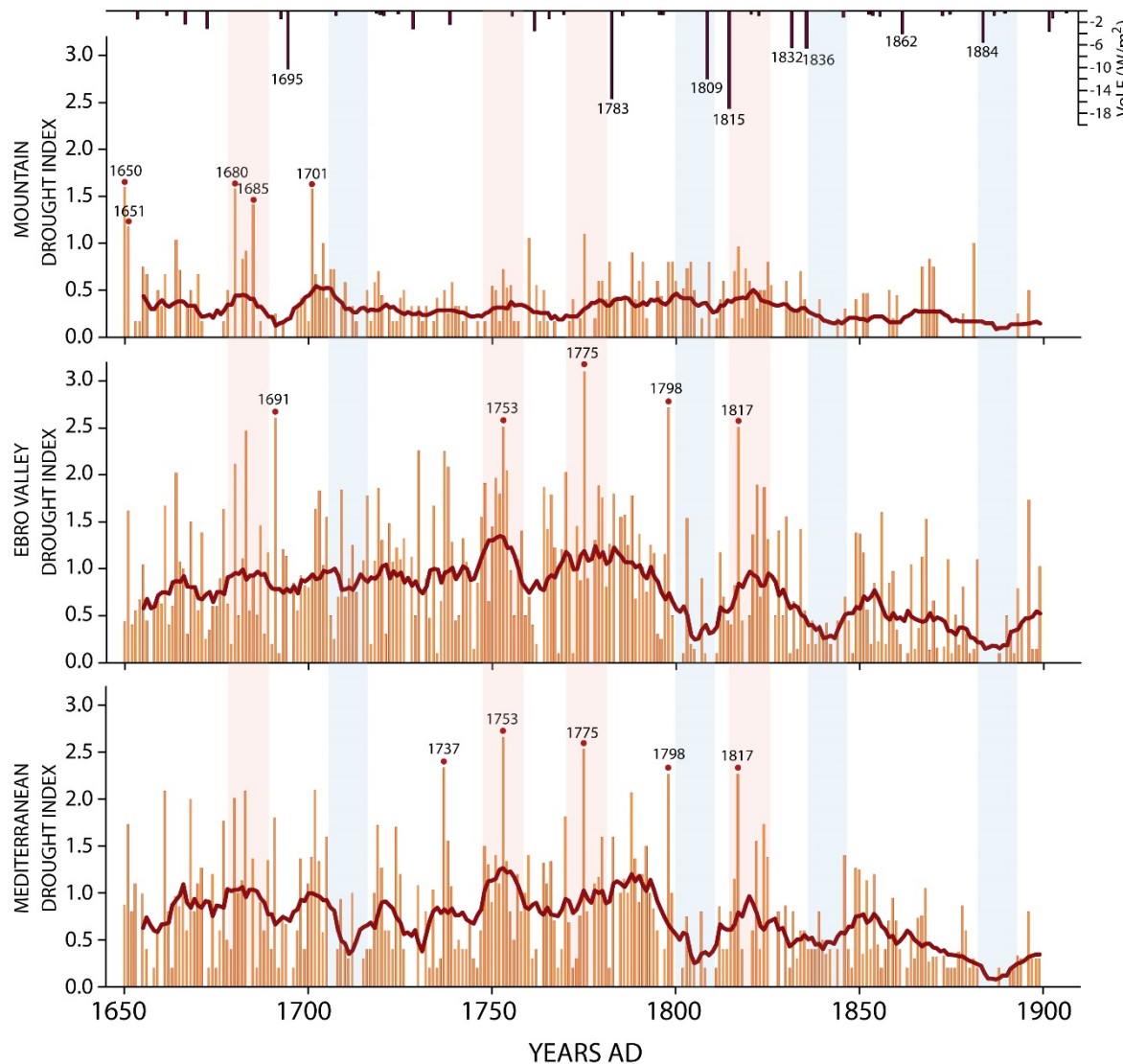


Figure 6. Drought indices of the three clusters, DIMOU (Mountain), DIEV (Ebro Valley)
and DIMED (Mediterranean). Vertical orange bars represent the drought index
magnitude, 0 denotes normal conditions, and 3 denotes an extreme drought year. The
extreme drought index years are also highlighted with a red circle. Extreme volcanic
events from Sigl et al., 2015, are shown in the top panel. Vertical pink shadows indicate
extreme common (for all three clusters) drought periods, while blue shadows indicate
common periods with fewer droughts.






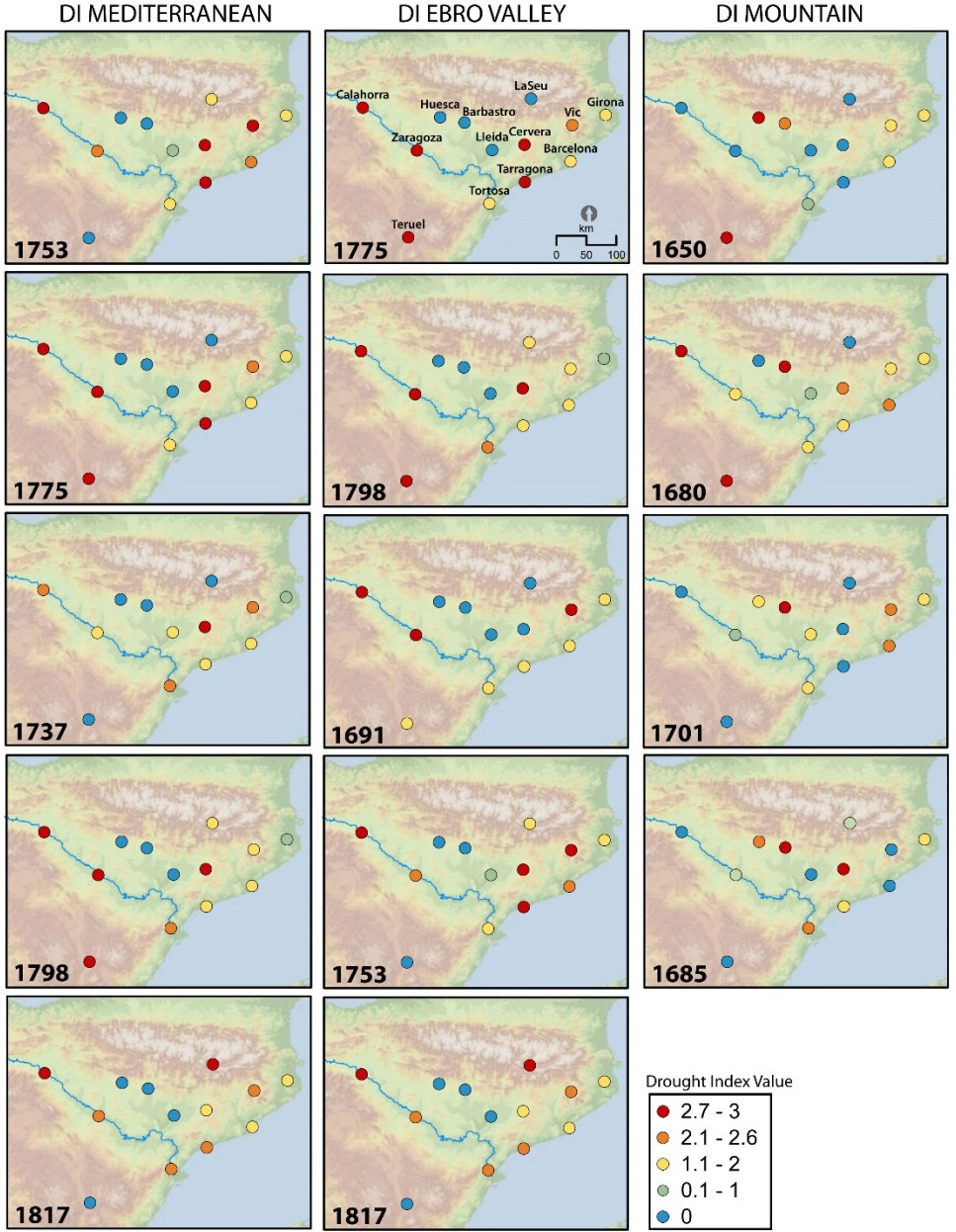

Figure 7. Spatial distribution of the most extreme drought years (based on the 99[th] percentile of the cluster drought indices). The distribution is ordered top-down. The drought index value (magnitude) for each site within the cluster is also represented. The legend of the drought index value is based on the 30[th], 60[th], 70[th] and 90[th] percentiles.

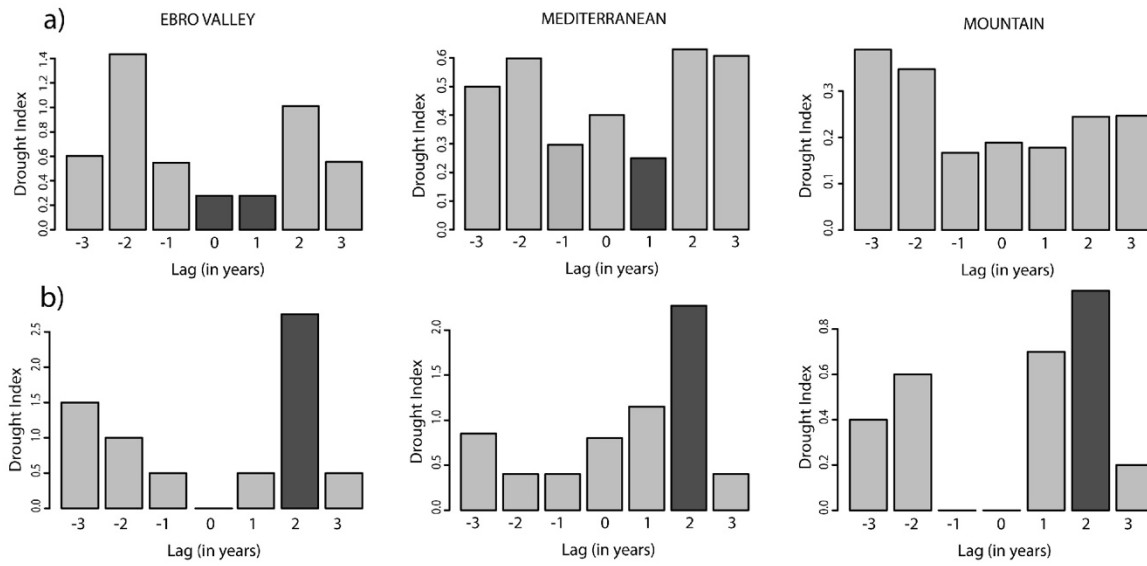


Figure 8. a) Superposed epoch analysis (SEA) of the three regional drought indices,
DIMOU (Mountain), DIEV (Ebro Valley) and DIMED (Mediterranean), with major volcanic
events from Sigl et al., 2015. Black shadows show significance at $p<0.01$, i.e., significantly
lower or higher drought index values after the volcanic event. b) SEA of only the
Tambora (1815) event showing a significant ($p<0.01$) increase in the drought index.








| Level | Type of ceremony |
|---|---|
| 0 | No ceremonies |
| 1 | Petition within the church |
| 2 | Masses and processions with the intercessor within the church |
| 3 | Pilgrimage to the intercessor of other sanctuary or church |




Table 3. Rogation levels according to the type of ceremony celebrated.

