# Peer review of "Rogation ceremonies: A key to understanding past drought"

_Climate of the Past, 2018_

## Referee Comment (RC1) · Anonymous Referee #1 · 7 Aug 2018

The paper aims to characterise the variability of droughts in NE Spain since 1650 using records from rogation ceremonies from 13 cities. This type of records have been used in the literature as proxy for droughts in the last years with success, as can be seen in the literature and is well reflected in the references of the manuscript. Most of those previous studies are focused on certain locations, but there have also been previous exercises analyzing jointly these records. The main novelty here is the use of cluster analysis to identify spatial patterns within NE using these rogation ceremonies. I have several major methodological problems in the type of treatment used in the manuscript that prevent me from acceptance. 1- For every location the authors generate and index which ranges from 0 to 3 depending on the frequency and type of rogations. According

to the manuscript, the index is computed as a weighted average of the reports found for a given year between December and August. The weight depends on the type of rogation held, according to a given protocol. In my view this must be interpreted with caution due to different reasons. First the same value can be reached with different extremes. Thus, a value of 2 (moderate drought) can be obtain with one single record of level 2 or with two records: one rogation ceremony of level one and another one of level 3 (1x1 + 1x3)/2=2. The climatic difference is really relevant, since in the second case the drought should have been much more extreme than in the first one. This index is semi quantitative because the levels are assigned after analyzing the ritual and due to the lack of overlap with the instrumental record, it is just an assessment expressed in a quantitative scale. Finally, the index is not linear, in the sense that a drought of level 3 should not necessarily be three times more intense than a drought of level 1. All these cautions should be taken into account when applying to the index built in the manuscript. The authors claim (l 249 for example), that they have obtained a continuous quantitative index, but these cautions are not mentioned in the text. 2-Next, a cluster analysis is performed to identify spatial patterns. According to the manuscript, there are three patterns: Mediterranean, Mountain and Ebro Valley. I think that this division does not make sense from the climatological viewpoint due to several reasons: - Lerida (other times called Lleida) and Cervera are two locations separated around 50 km, they are both included in the Ebro valley, at a similar distance from the sea and with no relevant mountains in between (see figure 1). On top, the pluviograms are very similar, check the Iberian climatologiacal Atlas, for instance (http://www.aemet.es/documentos/es/conocermas/recursos_en_linea/publicaciones_y_estudios/publicaciones/Atlas-climatologico/Atlas.pdf). However, Lerida is included within the Mountain cluster and Cervera within the Mediterranean one. This is difficult to understand. - Teruel, in the middle of the Iberian range, is included within the Mountain cluster, which is mostly composed by locations close to the Pyrenees. Teruel is around 400 km from the closest location in the cluster. Its precipitation regimen is poorly associated with those in the Pyrenees. Additionally, as can be seen in figure3, Teruel index is only

significantly correlated with Barbastro and non significantly with the rest. - Gerona (or Girona, depending on the text or figure) shows similar problems with the rest of the Mountain cluster with 3 nonsignificant correlations and two very poor correlations (up to 0.22). Anyone familiar with the climate of Spain (as the authors) should be aware of these issues, that are also evident in figure 5. Consequently, I think that of physical meaning of the cluster is very poor and the patterns might be an statistical artifact. This is not strange, since the usual clustering techniques use Euclidean distances to define clusters and they are appropriate for quantitative variables. Unfortunately, the methods section does not provide information on the distance used to measure the stations proximity. In my view, the authors should repeat the clustering process but applying a technique appropriate to their data (semiquantitative and nonlinear indices with a short range 0-3) and should interpret the results much more carefully. To add credibility to the exercise, I suggest that they compute the SPI or SPEI indices for the 13 locations during the instrumental period and check and compare the results with those obtained with the historical indices. This would provide a certain idea of the consistency of the results. Minor comments Language should be rechecked since there are several grammar errors The authors should unify terminology (Lleida/Lerida; Girona/Gerona) The references to gray literature in Spanish should be eliminated or minimized.

---

## Referee Comment (RC2) · Anonymous Referee #2 · 3 Sep 2018

This study is very interesting and provides new and valuable data to the scientific knowledge on droughts in the northeast of Spain in the last centuries. The main contribution to the historical climatology of this region lies in the fact that the study assembles an important set of series of rogation ceremonies, including two new unpublished series (Barbastro and Huesca). The study has potential to be published in Climate of the Past, however, in my opinion there are aspects of methodology and discussion that must be improved and completed in order to raise the overall quality of the article and achieve the quality standards of the journal.

Main remarks: 1. An important recommendation is about the presentation of the

method and its limitations. Data of rogation ceremonies were converted into a "Drought Index" (DI) which was developed and applied in previous publications, as referred by the authors. However, the DI description is not totally correct when the authors simply say that "rogation data was transformed into quantitative monthly series" since the DI is, in fact, defined by an ordinal scale of intensity of droughts. Therefore, the study is based in a semi quantitative approach (DI series), which must be clearly stated in the methodology, as also the inherent limitations for the significance of the DI series should be more detailed and emphasized (in section 2, "Methods"). 2. Another important weakness of the study is the total absence of information on the historical archives visited and basic description of sources gathered within the data collection. In text, I have found only a reference to the "Actas Capitulares" of the cathedrals. That's all? the single information provided on these important issues are the location and periods of the series (table I) which in my opinion is poor and quite insufficient to the readers and interested researchers. I suggest changing and complete this table or, preferably, add a new table with the recommended contents or even include a dedicated appendix. 3. In the methodology description the authors did not mention the completeness of the rogation records of the 13 collected series or even if there some possible gaps our periods with doubtful information from 1650 to 1899. Is it possible to estimate (approximately) the degree of temporal continuity of each series of rogation records? All uncertainties related with the study must be clearly stated. If the 13 series are complete permitting a suitable chronological analysis, please emphasize this fact, otherwise the readers may not be aware on the reliability of the data. 4. In the "Discussion" section some comments are missing about the apparent lack of coherence of cluster "Mountain" among the three defined drought patterns regions. As the authors pointed out, the correlations of DI within this group were weak or without statistical significance, but this evidence should be interpreted. What facts could explain this incoherency (or at least contribute to understand it). In my opinion these comments are relevant to support the consistency of the regional classification of drought series. 5. In the "Results" section is included a detection of the extreme drought years in the northeast of Spain

(3.3). Some aspects shown in figure 5 appear someway surprising, particularly when we compare the DI level occurred in quite closer cities in certain extreme drought years (see the example of Lleida and Cervera in 1775 and 1798) and some (apparently) contradictory results emerge. Since droughts are regional climatological events, not "local" phenomena, how can be explained such apparent spatial inconsistency? Some comments or plausible arguments should be added in Discussion section to avoid possible questions or doubts that, reasonably, may arise to the readers.

Minor comments: Line 129: "regional droughts" instead of regional drought"; Line 134: Consider replace "geological formations" by "geological units" or geological regions"; Table 1: add variables units (are totally absent); Cities names are not uniformized in the text, figures and tables (e.g. Lleida and Lerida, etc.)

———————————————————

---

## Author Comment (AC1) · 1 Oct 2018

Dear Editor, We very much appreciate the comments and suggestions made by the reviewers. Since both reviewers are concerned by the fact that we are using a quantitative approach with semi quantitative data, we will better explain the limitations of our proxy in the revised version (please see new version attached). We also clarify the nature of our data by performing an Empirical Cumulative Distribution Function. The derived drought indices can take values between 0 and 3 (see Fig. 2AB, included now in the manuscript), and thus can be considered as a continuous variable. In addition, as suggested by #Anonymous reviewer 1, we have now included a new paragraph in

the manuscript showing the 'validation' of our data, including the new Figure 5, which we believe clearly shows the strength of rogation ceremonies as drought proxies. Lines 240-252; 'To better understand the relationship with the derive drought indices and the instrumental series, we used the longest instrumental precipitation and temperature series covering the period 1786-2017 (Prohom et al., 2012; Prohom et al., 2015) for the city of Barcelona and thus overlap the rogation ceremony's period from 1786 to 1899. The instrumental series was homogenized and developed including data from cities nearby and along the Mediterranean coast (see Prohom et al., 2015 for details). We then calculated the Standardized Precipitation Index (SPI, McKee et a., 1993) and the Standardized Evapotranspiration and Precipitation Index (SPEI, Begueria et al., 2014) and calculated spearman correlation between DIMED and the SPI/SPEI at different time scales including a maximum lag of 12 months covering the period 1787-1899. To further explore the relationship between the drought indices inferred from historical documents and the instrumental drought indices through time, we performed 30 and 50 years moving correlations.' Lines 310-319; 'The maximum correlation (r=-0.53; p<0.001) between the Mediterranean Drought Index and the instrumental SPI over the full 113-year period (1787-1899 AD; Fig.5C) is found for the SPI of May with a lag of 4 months (SPIMAY_4 hereafter). Slightly lower, though still significant correlation, is obtained when using the SPEI of May with a lag of 4 months (SPEIMAY_4) (r=-0.50; p<0.001, Fig.5D). The moving correlations between SPIMAY_4 and DIMED for 30 and 50 years (Fig.5A; Fig.5B) show high and stable correlation through the full period. The relationship with the SPEIMAY_4 is also high and stable throughout the overlapping period, although lower than with SPIMAY_4.'

We performed a cluster analysis to study meaningful groups of historical documents that share common characteristics. We agree with the reviewers that multiple cluster techniques will provide different results, but in this specific case we believe that the three clusters have spatial coherency (as commented in detail in the point by point response below).

[Figure]

This is the Point-by-point response with which we respond to all suggestions and comments of the reviewers.

Anonymous reviewer 1. The paper aims to characterize the variability of droughts in NE Spain since 1650 using records from rogation ceremonies from 13 cities. This type of records have been used in the literature as proxy for droughts in the last years with success, as can be seen in the literature and is well reflected in the references of the manuscript. Most of those previous studies are focused on certain locations, but there have also been previous exercises analyzing jointly these records. The main novelty here is the use of cluster analysis to identify spatial patterns within NE using these rogation ceremonies.

Many thanks for the positive comments.

I have several major methodological problems in the type of treatment used in the manuscript that prevent me from acceptance. 1- For every location the authors generate and index which ranges from 0 to 3 depending on the frequency and type of rogations. According to the manuscript, the index is computed as a weighted average of the reports found for a given year between December and August.The weight depends on the type of rogation held, according to a given protocol. In my view this must be interpreted with caution due to different reasons. First the same value can be reached with different extremes. Thus, a value of 2 (moderate drought) can be obtain with one single record of level 2 or with two records: one rogation ceremony of level one and another one of level 3 (1x1 + 1x3)/2=2. The climatic difference is really relevant, since in the second case the drought should have been much more extreme than in the first one.

We appreciate the comments and understand the reviewer's concerns. We believe through the Empirical Cumulative Distribution Function analysis and the validation section we now assert the validity of the methods used to convert the categorical information to semiquantitative data. Please note that we have now changed quantitative by

semiquantitative throughout the whole manuscript. We further extend the explanation of the limitations of our data in. Lines 371-380; 'Further limitations of converting qualitative information into quantitative data refer to the fact that, for instance, a drought index of level 2 does not necessarily imply a drought twice as intense as a drought index of level 1. This is an inherent limitation when dealing with historical documents as a climate proxy, and different approaches have been applied in the scientific literature (Vicente-Serrano and Cuadrat, 2007; Dominguez-Castro et al., 2008). In our paper, we follow the methodology proposed in the Millennium Project (European Commission, IP 017008) and demonstrated in Domínguez-Castro et al., (2012)'. To that extent, the ECDF helped understanding the nature of the historical documents when transformed into semiquantitative data which confirm that they can be treated as a continuous variable'.

This index is semi quantitative because the levels are assigned after analyzing the ritual and due to the lack of overlap with the instrumental record, it is just an assessment expressed in a quantitative scale.

Please see answer above.

Finally, the index is not linear, in the sense that a drought of level 3 should not necessarily be three times more intense than a drought of level 1. All these cautions should be taken into account when applying to the index built in the manuscript. The authors claim (l 249 for example), that they have obtained a continuous quantitative index, but these cautions are not mentioned in the text.

We have now extended the description of the drought index limitations including the following suggested changes, please see above. However, the fact that the correlation of the overlapping period between the instrumental and the regional DIMED is very high and stable over time suggests that the rogations ceremonies can be considered as a drought indicator. In such a catholic society, similar droughts throughout the territory would trigger similar religious acts, which at the same time cost money. The authorities

and the church would not perform an expensive rogation ceremony of level 3, unless drought is severe, and the yearly harvest is in danger.

Next, a cluster analysis is performed to identify spatial patterns. According to the manuscript, there are three patterns: Mediterranean, Mountain and Ebro Valley. I think that this division does not make sense from the climatological viewpoint due to several reasons: - Lerida (other times called Lleida) and Cervera are two locations separated around 50 km, they are both included in the Ebro valley, at a similar distance from the sea and with no relevant mountains in between (see figure 1). On top, the pluviograms are very similar, check the Iberian climatologiacal Atlas, for instance (http://www.aemet.es/documentos/es/conocermas/recursos_en_linea/publicaciones_y_estudios/publicaciones/Atlas-climatologico/Atlas.pdf). However, Lerida is included within the Mountain cluster and Cervera within the Mediterranean one. This is difficult to understand. - Teruel, in the middle of the Iberian range, is included within the Mountain cluster, which is mostly composed by locations close to the Pyrenees. Teruel is around 400 km from the closest location in the cluster. Its precipitation regimen is poorly associated with those in the Pyrenees. Additionally, as can be seen in figure3, Teruel index is only significantly correlated with Barbastro and non significantly with the rest. - Gerona (or Girona, depending on the text or figure) shows similar problems with the rest of the Mountain cluster with 3 nonsignificant correlations and two very poor correlations (up to 0.22). Anyone familiar with the climate of Spain (as the authors) should be aware of these issues, that are also evident in figure 5. Consequently, I think that of physical meaning of the cluster is very poor and the patterns might be an statistical artifact.

We appreciate this a comment, and now have explained in the revised version. Lines 389-403; 'In addition, the clusters might not only be collecting climatic information but also diverse agricultural practices or even species. For instance, Cervera and Lleida, sharing similar annual precipitation totals, belong to the Mediterranean and the Mountain Drought Indices respectively. Lleida is located in a valley with an artificial irrigation system since the Muslim period, which is fed by the river Segre (one of the largest tributaries to the Ebro river). The drought in the Pyrenees is connected with a shortage of water for the production of energy in the mills as well as to satisfy irrigated agriculture. However, the irrigation system itself allowed them to manage the resource and resist much longer. Therefore, only the most severe droughts, and even so in an attenuated form, are perceived in the city. Cervera, located in the mountains, in the so-called pre-littoral system and its foothills, has a different precipitation dynamic more sensitive to the arrival of humid air from the Mediterranean. Besides, Lleida had a robust irrigation system that Cervera did not have. The droughts in Cervera are therefore more "Mediterranean" like and thus it is consistent its presence in the Mediterranean Drought Index.'

This is not strange, since the usual clustering techniques use Euclidean distances to define clusters and they are appropriate for quantitative variables. Unfortunately, the methods section does not provide information on the distance used to measure the stations proximity.

We apologize for that, although it was included in the submitted manuscript denoted as Figure 2. Now the cluster analysis is explained more clearly and in more detail. Lines 213-223; 'We used the Ward's method in which the proximity between two clusters is the magnitude by which the summed squared in their joint cluster will be greater than the combined summed square in these two clusters SS12−(SS1+SS2) (Ward, 1963; Everitt et al., 2001). Then, the root of the square difference between co-ordinates of pair of objects is computed with its Euclidian distance. Finally, for each cluster within the hierarchical clustering, quantities called p-values are calculated via multiscale bootstrap resampling (1000 times). Bootstrapping techniques does not require assumptions such as normality in original data (Efron, 1979) and thus represents a suitable approach applied to the semiquantitative characteristics of drought indices (DI) derived from historical documents. . .'

In my view, the authors should repeat the clustering process but applying a technique appropriate to their data (semiquantitative and nonlinear indices with a short range 0-3)

and should interpret the results much more carefully. To add credibility to the exercise, I suggest that they compute the SPI or SPEI indices for the 13 locations during the instrumental period and check and compare the results with those obtained with the historical indices. This would provide a certain idea of the consistency of the results.

We have now better justify the cluster technique applied. In addition, we have incorporated the validation section, including the calculation of the instrumental SPI and SPEI drought indices suggested by the reviewer. However, the validation between instrumental data and the Drought Indices derived from historical documents cannot be extended to the 13 cities due to the lack of overlapping periods. Most of the instrumental records in Spain, especially in small towns such as those studied here, begin in the second half of the 20th century. We believe that the high correlation found between the instrumental series of Barcelona and the Mediterranean Drought Index is already asserting the validity of our methodology to convert the rogation ceremonies into a continuous drought index.

Minor comments Language should be rechecked since there are several grammar errors. The authors should unify terminology (Lleida/Lerida; Girona/Gerona) The references to gray literature in Spanish should be eliminated or minimized.

Done.

**Anonymous reviewer 2. This study is very interesting and provides new and valuable data to the scientific knowledge on droughts in the northeast of Spain in the last centuries. The main contribution to the historical climatology of this region lies in the fact that the study assembles an important set of series of rogation ceremonies, including two new unpublished se- ries (Barbastro and Huesca). The study has potential to be published in Climate of the Past, however, in my opinion there are aspects of methodology and discussion that must be improved and completed in order to raise the overall quality of the article and achieve the quality standards of the journal.**

Many thanks for the positive comments.

Main remarks: 1. An important recommendation is about the presentation of the method and its limitations. Data of rogation ceremonies were converted into a "Drought Index" (DI) which was developed and applied in previous publications, as referred by the authors. However, the DI description is not totally correct when the authors simply say that "rogation data was transformed into quantitative monthly series" since the DI is, in fact, defined by an ordinal scale of intensity of droughts. Therefore, the study is based in a semi quantitative approach (DI series), which must be clearly stated in the methodology, as also the inherent limitations for the significance of the DI series should be more detailed and emphasized (in section 2, "Methods").

We appreciate the suggestions, which we believe have been now clarify. As responded in the general response. We also clarify the nature of our data by performing an Empirical Cumulative Distribution Function. The derived drought indices can take values between 0 and 3 (see Fig. 2AB, included now in the manuscript), and thus can be considered as a continuous variable. In addition, as suggested by #Anonymous reviewer 1, we have now included a new paragraph in the manuscript showing the 'validation' of our data, including the new Figure 5, which we believe clearly shows the strength of rogation ceremonies as drought proxies.

We performed the cluster analysis to study meaningful groups of historical documents that share common characteristics. We agree with the reviewers that multiple cluster techniques will provide different results, but in this specific case we believe that the three clusters have spatial coherency (as commented in detail in the point by point response below).

We have now extended the description of the drought index limitations including the following suggested changes. Lines 371-380;'Further limitations of converting qualitative information into quantitative data refer to the fact that, for instance, a drought index of level 2 does not necessarily imply a drought twice as intense as a drought index of level 1. This is an inherent limitation when dealing with historical documents as a climate proxy, and different approaches have been applied in the scientific literature

(Vicente-Serrano and Cuadrat, 2007; Dominguez-Castro et al., 2008). In our paper, we follow the methodology proposed in the Millennium Project (European Commission, IP 017008) and demonstrated in Domínguez-Castro et al., (2012). To that extent, the ECDF helped understanding the nature of the historical documents when transformed into semiquantitative data which confirm that they can be treated as a continuous variable..'

However, the fact that the correlation of the overlapping period between the instrumental and the regional DIMED is very high and stable over time suggests that the rogations ceremonies can be considered as a drought indicator. In such a catholic society, similar droughts throughout the territory would trigger similar religious acts, which at the same time cost money. The authorities and the church would not perform an expensive rogation ceremony of level 3, unless drought is severe and the yearly harvest is in danger.

2. Another important weakness of the study is the total absence of information on the historical archives visited and basic description of sources gathered within the data collection. In text, I have found only a reference to the "Actas Capitulares" of the cathedrals. That′s all? the single information provided on these important issues are the location and periods of the series (table I) which in my opinion is poor and quite insufficient to the readers and interested researchers. I suggest changing and complete this table or, preferably, add a new table with the recommended contents or even include a dedicated appendix.

We apologize for the absence of information on the historical archives that were visited. Now this information is included in the supplementary material Table S1.

3. In the methodology description the authors did not mention the completeness of the rogation records of the 13 collected series or even if there some possible gaps our periods with doubtful information from 1650 to 1899. Is it possible to estimate (approximately) the degree of temporal continuity of each series of rogation records? All

uncertainties related with the study must be clearly stated. If the 13 series are complete permitting a suitable chronological analysis, please emphasize this fact, otherwise the readers may not be aware on the reliability of the data.

We appreciate this comment. While the temporal length of each site was presented in Table 1, we provide more detailed information in the introduction and in the method sections. Lines 162-164; 'The extension of the consulted documents (described in Table S1) ranges from 461 years of continues data in Girona, to 120 years in Lleida, with an average of 311 years of data on each station.

4. In the "Discussion" section some comments are missing about the apparent lack of coherence of cluster "Mountain" among the three defined drought patterns regions. As the authors pointed out, the correlations of DI within this group were weak or without statistical significance, but this evidence should be interpreted. What facts could explain this incoherency (or at least contribute to understand it). In my opinion these comments are relevant to support the consistency of the regional classification of drought series.

We have now extended the discussion part. Lines 459-466; 'In particular, the mountain areas show less vulnerability to drought compared to the other regions. This is mainly due to the fact, that mountainous regions experience less evapotranspiration, more snow accumulation and convective conditions that lead to a higher frequency of thunderstorms during the summertime. In addition, the productive system of the mountain areas is not only based on agriculture but also on animal husbandry, giving them an additional source for living in case of extreme drought. This might explain the lower coherence among stations within the DIMOU.'

5. In the "Results" section is included a detection of the extreme drought years in the northeast of Spain (3.3). Some aspects shown in figure 5 appear someway surprising, particularly when we compare the DI level occurred in quite closer cities in certain extreme drought years (see the example of Lleida and Cervera in 1775 and 1798) and

some (apparently) contradictory results emerge. Since droughts are regional clima-tological events, not "local" phenomena, how can be explained such apparent spatial inconsistency? Some comments or plausible arguments should be added in Discussion section to avoid possible questions or doubts that, reasonably, may arise to the readers.

We appreciate the comments. We modified the corresponding text as follows: Lines 389-403; 'In addition, the clusters might not only be collecting climatic information but also diverse agricultural practices or even species. For instance, Cervera and Lleida, sharing similar annual precipitation totals, belong to the Mediterranean and the Mountain Drought Indices respectively. Lleida is located in a valley with an artificial irrigation system since the Muslim period, which is fed by the river Segre (one of the largest tributaries to the Ebro river). The drought in the Pyrenees is connected with a shortage of water for the production of energy in the mills as well as to satisfy irrigated agriculture. However, the irrigation system itself allowed them to manage the resource and resist much longer. Therefore, only the most severe droughts, and even so in an attenuated form, are perceived in the city. Cervera, located in the mountains, in the so-called pre-littoral system and its foothills, has a different precipitation dynamic more sensitive to the arrival of humid air from the Mediterranean. Besides, Lleida had a robust irrigation system that Cervera did not have. The droughts in Cervera are therefore more "Mediterranean" like and thus it is consistent its presence in the Mediterranean Drought Index.

Minor comments: Line 129: "regional droughts" instead of regional drought"; Line 134: Consider replace "geological formations" by "geological units" or geological regions"; Table 1: add variables units (are totally absent); Cities names are not uniformized in the text, figures and tables (e.g. Lleida and Lerida, etc.)

Done.

Figure 2. The empirical cumulative distribution function (ECDF) used to describe a

sample of observations for a given variable. Its value at a given point is equal to the proportion of observations from the sample that are less than or equal to that point. ECDF performed for the local drought indices (A) and the regional drought indices (B).

Figure 5. A) 30y moving correlation between DIMED and the instrumental computed SPI and SPEI. B) Same but 50y moving correlations. C) Correlation (Spearman) between DIMED and SPIMAY_4 for the full period (1787-1899). D) Correlation (Spearman) between DIMED and SPEIMAY_4 for the full period (1787-1899).

Please also note the supplement to this comment:
https://www.clim-past-discuss.net/cp-2018-67/cp-2018-67-AC1-supplement.zip

[Figure]

[Figure]

**Fig. 1.**

[Figure]

**Fig. 2.**

---

## Referee Report (RR1)

The manuscript is an interesting work on historical droughts in Northeast Iberian Peninsula, but in my opinion it needs some revisions before publishing:

1) In the literature you can find different definitions of the drought concept (atmospheric, meteorological, hydrological, agricultural), depending on the physical variable studied (relative humidity, rainfall, other elements of the hydrological cycle), and the duration of the event (days, months, seasons). I understand that here the authors are studying agricultural droughts, due to the origin of the data (rogations linked to agricultural production). In any case, it would be important to precise this point.

2) The nature of rogation ceremonies must be explained with more detail. For instance, is it possible to find a 'preventive rogation', that is, a ceremony organized before the event occur? In this sense, the date of the rogation is an important information. It may be the case that a dry winter provoked the rogation, but timely spring rainfalls yielded a good harvest. In that case, can we speak on 'drought'? In relation to previous comment, perhaps here we could speak on dry winter (meteorological drought), but not on 'agricultural drought', and, in consequence, this event is not comparable with other characterized by the water deficit during an entire year. The 'annual' index (from December to August) may mask important intra-annual fluctuations, in my opinion it is preferable to divide the information into seasonal indices, following the different phases of the plant growing, from seed (autumn) to harvest (summer). In addition, all rogations are linked to cereal production? Other plants (fruits, olive trees) have different climatic limitations, and it would be possible that a single meteorological event (for instance, dry spring) was harmful for a specific plant, but not for another (for instance, the barley is more tolerant to drought than wheat).

3) I have doubts on the classification of the rogations (lines 205-207, Table 2). Were the ceremonies the same in all the cities and during the whole time period, from 1650 to 1899? Severity indices are based on the type of ceremony, but is it a reliable criteria? In the discussion (lines 404-411), authors say that 'an index of level 2 does not necessarily imply that a drought was twice as intense as a drought classified as level 1, nor that the change in the intensity of droughts from level 1 to level 2 or from level 2 to level 3 has to be necessarily equivalent'. In that case, how must we interpret these indices? In my opinion, these indices only specify the nature of the ceremonies organized as response to natural hazards, but do not inform on the severity of the climatic event. In consequence, what is their utility from a climatic point of view? In my opinion, the binomial distribution (occurrence or not) is the more appropriate statistical approach to the treatment of this information.

4) Clustering is an appropriate tool to classify and group local series into regional series. There are very different clustering algorithms, hierarchical and not hierarchical. Why have you used Ward method with Euclidean distance, and not, for example, the non-hierarchical k-means, or other methods as the principal component analysis? Results of clustering must yield groups more or less

homogeneous, but the chosen number of clusters is normally arbitrary. Why do you distinguish between Mediterranean and Ebro Valley group (dendrogram, Figure 3), if, as you say (lines 450-451) 'the high correlation between DIEV and DIMED is suggesting similar climatic characteristics'?

5) Validation of the regional drought indices is made using the overlapping period 1786-1899 between documentary and instrumental data. But, as you say in the discussion (lines 390-392) 'the apparent low frequency of rogations in the 19th century could be explained by a combination of political instability, and the loss of religiosity and historical documents'. I would add changes in the socioeconomic structures, organization of the cereal production, agricultural techniques, etc. In consequence, this period is not valid to calibrate and/or validate the rogation series in previous centuries. The cultural background, economic organization and technology of the 19th century was not the same that in previous centuries, and calibrations established for 19th century are not applicable to 17th century! In fact, you do not use this calibration (regression in Figure 5) to interpret previous data, only to validate the index during the 19th century. Besides, this analysis is only made for DIMED (Barcelona), and not for the other points in the studied area. I suggest to remove this analysis (and the Figure 5).

6) Superposed epoch analysis (SEA). Although you give a reference, a brief explanation on the basis and procedures of this method would be important.

7) Minor questions:

Line 301:'The cluster analysis (CA, see methods) using the DI of the 13 locations for the period of 1650-1899 AD revealed three significantly coherent areas…' Erratum, I suppose, clustering is made using the period 1650-1770, common to all the stations, although the classification obtained is after applied to the complete period until 1899.

Lines 373-375: 'However, two years after the Tambora eruption in April 1815, there was a significant ($p<0.05$) increase in the three drought indices…' However? The time life of volcanic aerosols in the atmosphere is around one to two years. The Tambora aerosols caused a radiative forcing of the global climate system of about 5.6W/m$^2$ for one to two years following the eruption (Brönnimann and Krämer, 2016). In consequence, this increase in the drought indices may be caused not by the volcanic eruption, but by the return to 'normal' conditions (or not forced climate variability).

Lines 417-418: 'the local series are compared with the regional reference series as a basic element of quality control'. But, if the regional series is obtained from the average of local series, here we have a circularity problem.

Line 432: 'the local series are separated by tens or hundreds of kilometers'. If you speak on meteorological droughts, this is not a problem, because the dynamical

conditions provoking dry conditions are associated to the predominance of anticyclonic conditions, and the spatial extension of an anticyclone may be much greater. Again, we are speaking on the drought concept. Meteorological, hydrological, agricultural?

Figure 5. Significance level must be added in the figures. D), E), F), correlation is the Pearson correlation coefficient?

Figure 7.The legend is arbitrary, why do you distinguish between 2.1-2.6 and 2.7-3 DI values?

Reference

Brönnimann S, Krämer D. 2016. Tambora and the 'Year Without a Summer' of 1816. A perspective on Earth and Human Systems Science. Geographica Bernensia G90, 48 pp, doi:10.4480/GB2016.690.01

---

## Editor Decision (ED1)

**Reviewer 1:**

I think that the authors have addressed only part of my previous concerns, but the core of the problems that I find have not been solved. Specially, I am not convinced by the application of the clustering technique to the indices generated by the authors. The indices are semi quantitative, continuous and nonlinear variables. I think that, under these conditions, the use of the cluster techniques is questionable. In the new version I do not find convincing arguments justifying this application. The inconsistency of the results reinforces this. Now, the authors emphasize the limitations in the DI Mountain cluster, which is Ok. However, I still think that it is a purely statistical artifact. I have provided different reasons I my previous reviews. On top of them, I provide an additional one. Figure 7 shows the spatial patterns of extreme drought years. 1685 and 1701 (bottom right panels of the figure) show that in these 'extreme' years of the mountain cluster, Teruel shows an index value of 0. I think this is not acceptable and is another proof of the lack of physical foundation of this cluster analysis in an area of high precipitation and temperature variability, as acknowledged by the authors in lines 647-658.

The new version of the discussion is focused in justifying that rogation records are good proxies for droughts. This is has been proven in the literature and I am not questioning at all their value as local indicators of droughts. My point is on how the cluster is applied and interpreted in that specific case, not on the validity of rogations.

Panel a) of figure 8 is from my viewpoint another proof of lack of consistency. How can the authors explain that they find in the Mediterranean cluster a significant signal before the volcanic eruptions? (year -1) Do they have an explanation for this?. I think that the only robust result is the impact of the Tambora eruption in panel b, a result previously reported in papers co-authored by some of the authors.

So, I do not think the paper is acceptable for publication.

Additional comments

- The authors claim that they have computed the SPEI index since 1787. This index requires instrumental data to compute the AED. Which data have they used? This should be explained in the text, since, as they claim, the only long instrumental series in the region is the Barcelona temperature series. Is this a combination of instrumental and proxy records?. If so, this must be carefully described in the text.

- The language needs further revision. I asked in my previous review but it does not seem to have been taken into account too seriously.

- Some references are missing or wrong. Examples:

line 168 AEMET 2012 missing

line 744-745 I have not been able to find Dominguez-Castro and Garcia Herrera GRL 2016

The reference to Garcia-Ruiz 2001 about climate change impacts in the Mediterranean should be updated. See for example

The climate of the Mediterranean region: from the past to the future

2012 , Elsevier Insights, 592pp, ISBN: 978-0-12-416042-2 , Ed. Lionello P.

Mediterranean Climate Variability

2006, Elsevier, Amsterdam, ISBN: 0-444-52170-4, 438 pp, Eds: Lionello P., P. Malanotte-Rizzoli and R. Boscolo

- DIMED appears in l 272, while its meaning is explained in line 324

**Reviewer 2 (new):**

The manuscript is an interesting work on historical droughts in Northeast Iberian Peninsula, but in my opinion it needs some revisions before publishing:

1) In the literature you can find different definitions of the drought concept (atmospheric, meteorological, hydrological, agricultural), depending on the physical variable studied (relative humidity, rainfall, other elements of the hydrological cycle), and the duration of the event (days, months, seasons). I understand that here the authors are studying agricultural droughts, due to the origin of the data (rogations linked to agricultural production). In any case, it would be important to precise this point.

2) The nature of rogation ceremonies must be explained with more detail. For instance, is it possible to find a 'preventive rogation', that is, a ceremony organized before the event occur? In this sense, the date of the rogation is an important information. It may be the case that a dry winter provoked the rogation, but timely spring rainfalls yielded a good harvest. In that case, can we speak on 'drought'? In relation to previous comment, perhaps here we could speak on dry winter (meteorological drought), but not on 'agricultural drought', and, in consequence, this event is not comparable with other characterized by the water deficit during an entire year. The 'annual' index (from December to August) may mask important intra-annual fluctuations, in my opinion it is preferable to divide the information into seasonal indices, following the different phases of the plant growing, from seed (autumn) to harvest (summer). In addition, all rogations are linked to cereal production? Other plants (fruits, olive trees) have different climatic limitations, and it would be possible that a single meteorological event (for instance, dry spring) was harmful for a specific plant, but not for another (for instance, the barley is more tolerant to drought than wheat).

3) I have doubts on the classification of the rogations (lines 205-207, Table 2). Were the ceremonies the same in all the cities and during the whole time period, from 1650 to 1899? Severity indices are based on the type of ceremony, but is it a reliable criteria? In the discussion (lines 404-411), authors say that 'an index of level 2 does not necessarily imply that a drought was twice as intense as a drought classified as level 1, nor that the change in the intensity of droughts from level 1 to level 2 or from level 2 to level 3 has to be necessarily equivalent'. In that case, how must we interpret these indices? In my opinion, these indices only specify the nature of the ceremonies organized as response to natural hazards, but do not inform on the severity of the climatic event. In consequence, what is their utility from a climatic point of view? In my opinion, the binomial distribution (occurrence or not) is the more appropriate statistical approach to the treatment of this information.

4) Clustering is an appropriate tool to classify and group local series into regional series. There are very different clustering algorithms, hierarchical and not hierarchical. Why have you used Ward method with Euclidean distance, and not, for example, the non-hierarchical k-means, or other methods as the principal component analysis? Results of clustering must yield groups more or less homogeneous, but the chosen number of clusters is normally arbitrary. Why do you distinguish between Mediterranean and Ebro Valley group (dendrogram, Figure 3), if, as you say (lines 450-451) 'the high correlation between DIEV and DIMED is suggesting similar climatic characteristics'?

5) Validation of the regional drought indices is made using the overlapping period 1786-1899 between documentary and instrumental data. But, as you say in the discussion (lines 390-392) 'the apparent low frequency of rogations in the 19th century could be explained by a combination of political instability, and the loss of religiosity and historical documents'. I would add changes in the socioeconomic structures, organization of the cereal production, agricultural techniques, etc. In consequence, this period is not valid to calibrate and/or validate the rogation series in previous centuries. The cultural background, economic organization and technology of the 19th century was not the same that in previous centuries, and calibrations established for 19th century are not applicable to 17th century! In fact, you do not use this calibration (regression in Figure 5) to interpret previous data, only to validate the index during the 19th century. Besides, this analysis is only made for DIMED

(Barcelona), and not for the other points in the studied area. I suggest to remove this analysis (and the Figure 5).

6) Superposed epoch analysis (SEA). Although you give a reference, a brief explanation on the basis and procedures of this method would be important.

7) Minor questions:

Line 301:'The cluster analysis (CA, see methods) using the DI of the 13 locations for the period of 1650-1899 AD revealed three significantly coherent areas…' Erratum, I suppose, clustering is made using the period 1650-1770, common to all the stations, although the classification obtained is after applied to the complete period until 1899.
Lines 373-375: 'However, two years after the Tambora eruption in April 1815, there was a significant ($p<0.05$) increase in the three drought indices…' However? The time life of volcanic aerosols in the atmosphere is around one to two years. The Tambora aerosols caused a radiative forcing of the global climate system of about 5.6W/m2 for one to two years following the eruption (Brönnimann and Krämer, 2016). In consequence, this increase in the drought indices may be caused not by the volcanic eruption, but by the return to 'normal' conditions (or not forced climate variability).
Lines 417-418: 'the local series are compared with the regional reference series as a basic element of quality control'. But, if the regional series is obtained from the average of local series, here we have a circularity problem.
Line 432: 'the local series are separated by tens or hundreds of kilometers'. If you speak on meteorological droughts, this is not a problem, because the dynamical conditions provoking dry conditions are associated to the predominance of anticyclonic conditions, and the spatial extension of an anticyclone may be much greater. Again, we are speaking on the drought concept. Meteorological, hydrological, agricultural?
Figure 5. Significance level must be added in the figures. D), E), F), correlation is the Pearson correlation coefficient?
Figure 7.The legend is arbitrary, why do you distinguish between 2.1-2.6 and 2.7-3 DI values?
Reference
Brönnimann S, Krämer D. 2016. Tambora and the 'Year Without a Summer' of 1816. A perspective on Earth and Human Systems Science. Geographica Bernensia G90, 48 pp,

doi:10.4480/GB2016.690.01

**Reviewer 3 (new)**

This study compiles and quantitatively analyses drought indices derived from documentary data on rogation ceremonies in northeastern Spain to further insights into historical droughts but also to understand the role of volcanic forcing on past event.

The most importantly contribution of the paper is that it provides a very interesting insight into both the strengths and weaknesses of documentary sources, especially approaches that seek to derive quantitative estimates from qualitative data. It is my view that the paper should be accepted following minor revision. I suggest some points below that that the authors need to address. Most of my comments seek clarity and explanation. I also request the authors to give the paper a thorough edit. There are instances of misspelling or improper English scattered throughout the paper. This deserves considerable attention. There are also instances of long tracts of text that make it hard for the reader to track key points. Please use paragraphs more effectively to deal with this. There are also very long sentences at times (e.g. end of introduction) that need to be broken up.

I would like to know more about the documentary sources consulted. We don't get to know much about this aspect despite so much interpretation later depending on these sources. In line with comments from previous reviewers the original sources and their consultation/analysis needs to be given greater attention in the paper.

The key methodological steps in the paper are as follows:
1) Development of a semi-quantitative series from qualitative data derived from documentary sources on rogation ceremonies. This is done using an established technique. I have no concerns to note.
2) Clustering of series to develop regional drought indices
This again seems to follow best practice. Importantly the analysis is not entirely statistically based and physical reasoning around the derived clusters is given. This is important as such techniques are somewhat subjective and the authors are transparent in their choices. The limitations of the approach are clearly articulated in the discussion.
3) Validation of the resultant series against instrumental records.
4) The performance of an epoch analysis to detect volcanic influence on historical droughts. This section is given the least attention in methods and most prominence in the abstract. I think that the authors need to explain this approach in more and sufficient detail to allow reader fully understand what they are doing here. A short paragraph should suffice. Why this method and what are the assumptions/strengths/weaknesses around the approach and desired attribution statements.

I do not have local knowledge of the region but I find the results interesting, especially for the mountainous region. It seems the other two regions show similar results that are coherent with findings from previous studies. Indeed in discussion this aspect of the coherence of results needs to be moved further up. This is important information to have before getting into the limitations.

I find the weaker results from the mountain region interesting. Some effort is expended on trying to explain why this difference and at times the authors get into attributing different processes. First, is this something seen in recent times when we have measurements? Second, given that the performance of a rogation was done based in part on the wishes of agricultural institutions, is there a risk that mountainous regions would have weaker political power in influencing rogations. Therefore, the lack of intense droughts in mountainous region or the disagreement with other regions, may be due to its weaker economic importance rather than anything to do with drought directly? I think the authors need to mention this possibility.

I also think that the authors could make more of the issues they run into for the mountainous region as a case of the challenges of using documentary sources. This needs to be mentioned in the abstract as its lessons are important for other studies.

The authors rightly state that the drought index for the mountainous region should be treated with extreme caution.

Need to be careful of tense used in abstract.

---

## Author Response (AR2)

Dear Editor, this is the point by point response to the reviewer's comments.

**Reviewer.**
**I think that the authors have addressed some of the questions raised in my previous review and I recognize this effort. However, I think that some of my concerns have not been properly addressed.**
**- Physical meaning of the clusters. I think that the authors fail in providing a convincing explanation for their proposed clusters. In their reply they acknowledge that the differences between Lleida and Cervera are not due to climatic factors, but rather to the higher resilience of Lleida to precipitation shortage due to the irrigation system. Thus, rogations in Lleida should only detect the most extreme events and their sensitivity to drought is different. However this different sensibility to water shortage is not taken into account in the clustering, when all the series are treated equally.**

We appreciate your previous comments that helped enhancing the quality of the manuscript. In the new version of the manuscript we have further addressed and re-organized the discussion to show the potential pitfalls of the cluster analysis and the use of rogation ceremonies as a climatic proxy. Please see new section from lines 393 to 512.
We have now included a new paragraph explaining the drawbacks when dealing with historical documents and clusters (now 437-484). We clarify your line "differences between Lleida and Cervera are not only due to climatic factors'. Rather we say 'the clusters might not only be collecting climatic information but also diverse agricultural practices or even species', which is considerably different.
All proxy records used to infer past climate conditions contain non-climatic information. For example, in tree-rings records, which are extensively used to develop reconstructions, climate explains between 40 to 60% of the tree-ring growth variance (in the best cases). This does not invalidate the proxy as a paleoclimatic source, but it adds a range of uncertainties and feasible hypothesis to elucidate that remaining unexplained percentage of variance. Here, the fact that Lleida is not included in the Mediterranean cluster, is precisely reflecting not only slightly different climatic conditions, but also a distinctive response of its population to drought events. As you point out in your comment, it might be only showing the most extreme events due to a higher resilience to droughts (because of their irrigation system), and that is why is indeed included in the Mountain cluster and not in the Mediterranean one. We hope that the new paragraphs included in the discussion would help addressing the different limitations of the use of rogation data as a climatic proxy.

**A physically consistent explanation of the inclusion of the Teruel series in the Pyrenees cluster is not provided in the reply. According to figure 4, Teruel is only significantly correlated with Huesca, but the explained variance is lower that 4%. No other significant correlations are found for Teruel within this cluster.**

In fact, we talk about a "Mountain cluster", which includes towns located within a higher elevation or latitude, such is the case of Teruel. This city is located at 915 m.a.s.l. being one of the highest capitals in Europe. Although it is located south of the rest of the cities within the DIMOU cluster, its agreement with the general DIMOU (r=0.55, p<0.05) (as a basic quality control method) denotes that there is a coherent and common shared regional signal. All the DI local stations included in DIMOU show significant correlations with the regional DIMOU, suggesting that there are particular regional scale events. However, we agree that the climatic interpretation of the DIMOU cluster should be treated with caution.
We have now included these and other potential drawbacks in lines 437 to 484.

**Consequently, I still think that the clusters have no climatological support, which rests validity to the rest of the analysis.**

We agree partially with this comment. Two of the three clusters analyzed here (DIEV and DIMOU) have a strong climatological support. The additional quality control approaches (calibration/ verification and relationship between series) are likely to be robust enough for their interpretation.  In addition, such results can be of special interest since DIEV and DIMOU represent geographical areas where other climate proxies (like tree-rings) are scarce.
We agree with specific limitations of the *pro-pluviam* rogations ceremonies as a climatic (drought) proxy in the Mountain cluster and we tried to explicitly expose such limitations in the new discussion section.

**Validation of the series. I acknowledge the effort in validating the series with the inclusion of the Barcelona instrumental series, which shows an overlapping period with the rogation series. However, I do not think that the authors have provided a fair comparison since they are comparing a local series (Barcelona), with a Regional average (DIMED). This has several problems which have been treated in model verification (see for instance Gilleland et al 2009 and references therein). Since the Barcelona rogation series is available, they should provide the comparison of the two local series during the overlapping period if they want to provide a convincing validation. So, according to these points, the clustering, which is the most original part of the paper, should be reconsidered to get a partition based on the climatic signal in a convincing way. On the other hand a fair validation should be provided.**

We have now included DIBCARCELONA as requested to complete our analysis (see new Figure 5) and further discussed the calibration/ verification approach (see lines 485-512).

**Minor point**
**Language is still an issue and should be checked carefully. For example line 193 of the new version: '461 years of continues data'**

Done.

[revised manuscript text omitted]

---

## Author Response (AR3)

**Dear editor, this is the point by point response with which we aim to respond to all the questions raised by the reviewers.**

**Reviewer 1.**
I think that the authors have addressed only part of my previous concerns, but the core of the problems that I find have not been solved. Specially, I am not convinced by the application of the clustering technique to the indices generated by the authors.
Sorry for the misunderstanding, we did not apply the clustering technique to compute the indices but to evaluate similarities among local series. We clarify this in lines 231-233.

The indices are semi quantitative, continuous and nonlinear variables. I think that, under these conditions, the use of the cluster techniques is questionable. In the new version I do not find convincing arguments justifying this application. The inconsistency of the results reinforces this.
We do not agree with the reviewer on this point. We think that our results are not inconsistent. The drought indices inferred from low elevation sites show common patterns suggesting common climatic driving forces. In addition, the mountain environment drought indices show more local patterns and do not reflect as well the temporal patterns of the low elevation indices. The limitations on the use of rogations as climatic proxies is not related to the technique of the analysis as rogations in different environments reflect climate events that occur at different time scales.

Now, the authors emphasize the limitations in the DI Mountain cluster, whihc is Ok. However, I still think that it is a purely statistical artifact. I have provided different reasons I my previous reviews. On top of them, I provide an additional one. Figure 7 shows the spatial patterns of extreme drought years. 1685 and 1701 (bottom right panels of the figure) show that in these 'extreme' years of the mountain cluster, Teruel shows an index value of 0.
What is meant is that 1685 and 1701 cannot be considered as extreme drought years in Teruel. From the regional tree-ring width hydroclimate reconstruction near Teruel (Tejedor et al., 2016) we can see that both years were considered as 'normal'. This also means that the DI Mountain has a weaker and less robust physical meaning. We clearly stated this in the discussion and conclusion sections.

I think this is not acceptable and is another proof of the lack of physical foundation of this cluster analysis in an area of high precipitation and temperature variability, as acknowledged by the authors in lines 647-658.

We think we have detailed the potential drawbacks of the DI Mountain and the strengths and weaknesses of performing clustering techniques. Despite Teruel's high elevation (900 m.asl), the annual mean precipitation is fairly low (378 mm; AEMET, 2019). Nevertheless, the monthly precipitation varies with respect to that in the Ebro Valley, driven by convective storms during August, September and October. The strong precipitation and temperature variability are reflected in the individual DI clustered in the DI Mountain, including a weaker or less robust signal as opposed to the solid and neater DI Mediterranean and DI Ebro Valley (Figures 4 and 6). We do not see it as a lack of physical foundation of the cluster but as a reflection that rogation ceremonies in high-elevation environments had a greater variability and intensity through time than in the low lands and Mediterranean coast. Therefore, DI Mountain results must be treated with caution, as we stated in the manuscript, but that does not invalidate it.

The new version of the discussion is focused in justifying that rogation records are good proxies for droughts. This is has been proven in the literature and I am not questioning at all their value as local indicators of droughts.
We also demonstrate that in some areas (Ebro Valley and Mediterranean coast) rogation records are suitable proxies for droughts not only as local indicators but also at a regional scale, which we believe is a novel finding.

My point is on how the cluster is applied and interpreted in that specific case, not on the validity of rogations.
We did not apply the clustering technique to compute the indices but to evaluate similarities among local series (lines 231-233).

Panel a) of figure 8 is from my viewpoint another proof of lack of consistency. How can the authors explain that they find in the Mediterranean cluster a significant signal before the volcanic eruptions? (year -1) Do they have an explanation for this?.
There are still uncertainties on the dating of volcanic events from ice-cores. Here, we only include the largest events on ice-cores from Sigl et al., 2015, which improve the uncertainty of the dating with respect to Gao et al., 2008 or Crowley and Utterman, 2013. However, there are still events with dating issues of +- 1 years. Therefore, we have been more restrictive with the significant level, now adjusted to $p<0.01$.

[Figure]

I think that the only robust result is the impact of the Tambora eruption in panel b, a result previously reported in papers co-authored by some of the authors.

Although is true that a drought response was also found in Dominguez-Castro et al., 2012 after the Tambora event, the methodology, the sites and the extend of the results are different. The response to Tambora event is large and significant in the three indices, including the DI Mountain, thus reaffirming their validity as extreme droughts proxies, and at the same time proving the consistency of these results. Lines 598-617.

So, I do not think the paper is acceptable for publication.
Additional comments
-      The authors claim that they have computed the SPEI index since 1787. This index requires instrumental data to compute the AED. Which data have they used? This should be explained in the text, since, as they claim, the only long instrumental series in the region is the Barcelona temperature series. Is this a combination of instrumental and proxy records?. If so, this must be carefully described in the text.

We calculated the SPEI with the SPEI package (Begueria et al., 2014). From the various ways of calculating evapotranspiration we chose Thornwaite, which requires only temperature and latitude as inputs. These details are included in lines 275-277.
We used the temperature and precipitation instrumental data from Prohom et al., 2012; 2015 that can be downloaded here;
http://www.meteo.cat/wpweb/climatologia/serveis-i-dades-climatiques/serie-climatica-historica-de-barcelona/

-      The language needs further revision. I asked in my previous review but it does not seem to have been taken into account too seriously.
This has been completed. The manuscript was reviewed by a native English speaker.
-      Some references are missing or wrong. Examples:
line 168 AEMET 2012 missing
Corrected.

line 744-745 I have not been able to find Dominguez-Castro and Garcia Herrera GRL 2016
Corrected.

The reference to Garcia-Ruiz 2001 about climate change impacts in the Medittarean should be updated. See for example
The climate of the Mediterranean region: from the past to the future
, Elsevier Insights, 592pp, ISBN: 978-0-12-416042-2 , Ed. Lionello P.
Mediterranean Climate Variability
Done.

2006, Elsevier, Amsterdam, ISBN: 0-444-52170-4, 438 pp, Eds: Lionello P., P. Malanotte-Rizzoli and R. Boscolo
-      DIMED appears in l 272, while its meaning is explained in line 324
Corrected.

**Reviewer 3.- Reconsidered after major revisions**

The manuscript is an interesting work on historical droughts in Northeast Iberian Peninsula, but in my opinion it needs some revisions before publishing:

Thank you for your comments.

1) In the literature you can find different definitions of the drought concept (atmospheric, meteorological, hydrological, agricultural), depending on the physical variable studied (relative humidity, rainfall, other elements of the hydrological cycle), and the duration of the event (days, months, seasons). I understand that here the authors are studying agricultural droughts, due to the origin of the data (rogations linked to agricultural production). In any case, it would be important to precise this point.

We appreciate your comment. However, it is difficult to determine one single type of drought. In cereals of long cycle, without any other artificial support, during the period of the year in which the harvest is in progress both the meteorological and the agricultural drought converge. The rogation was not only agronomical or focused on a drought or agricultural problem. They already inferred that the problem was meteorological. And they always asked for timely rain, appropriate rain, or consistent rain. Sometimes in despair they even asked that at least there be spray! In other words, they asked for the occurrence of a meteorological phenomenon. In consequence, the follow-up or sentinel that gives them information is agricultural, but their answer is by a meteorological anomaly, and they ask for the development of a normalized meteorology, that in consequence will allow a development of the appropriate agriculture. (Lines 199-206).

2) The nature of rogation ceremonies must be explained with more detail. For instance, is it possible to find a 'preventive rogation', that is, a ceremony organized before the event occur? In this sense, the date of the rogation is an important information. It may be the case that a dry winter provoked the rogation, but timely spring rainfalls yielded a good harvest. In that case, can we speak on 'drought'? In relation to previous comment, perhaps here we could speak on dry winter (meteorological drought), but not on 'agricultural drought', and, in consequence, this event is not comparable with other characterized by the water deficit during an entire year.

Rogation ceremonies are a relatively new climatic proxy indicator. Most of the methodological details and criteria can be found in Martín-Vide & Barriendos (1995), and Barriendos (1997). "Preventive" rogation is a concept that indicates a first institutional response when drought is in an early stage of crop affectation meaning that; i) no damages are recorded, ii) cereal crop is showing first negative effects by drought, iii) any rainfall event can correct the drought.
Of course, a systematic record of dates for this and other rogation levels is important, in order to consider severity and duration of drought events.

Drought is a cumulative phenomenon. Then, as you comment, certainly a dry winter can give any "preventive" rogation of level 1 in early spring. In fact, drought in winter shows effects in agriculture during the next months. This affectation is recorded by the rogation system. Then, if drought is persistent during the following spring, rogation ceremonies convoked by institutions are increased to higher levels.

To answer your interesting question, if we would work just for economic or agricultural history, the truth is that this short drought would not exist, or it will not be considered because of absence of impact. However, in the context of climatic variability, any minor rogation convoked for a short event, even without impacts on agriculture, is recording a meteorological drought. In a similar way, instrumental records can show months or seasons with low values of precipitation, but cumulative yearly value can be normal if other months are rainier. If rogation proxies can detect a high-resolution event, we think this is an improvement on knowledge of climatic variability at different time scales.

Finally, we agree that the rogations system is a proxy developed within an agricultural framework (institutions monitoring crops) and thus the information available is in fact related to meteorological and agricultural droughts.

The 'annual' index (from December to August) may mask important intra-annual fluctuations, in my opinion it is preferable to divide the information into seasonal indices, following the different phases of the plant growing, from seed (autumn) to harvest (summer). In addition, all rogations are linked to cereal production? Other plants (fruits, olive trees) have different climatic limitations, and it would be possible that a single meteorological event (for instance, dry spring) was harmful for a specific plant, but not for another (for instance, the barley is more tolerant to drought than wheat).

This is a good remark. However, within the study area the main agricultural cereals are wheat and barley and are generally mixed i.e., cereal plots are mixed, and farmers usually do not focus on one single cereal but grow several. Therefore, in the study area there are not extensive areas that we can identified as of a single cereal. The greater agricultural homogeneity found in the Mediterranean and Ebro Valley might be one of the reasons of finding a common pattern. The greater specificity in mountain environment crops may be one of the causes of a weaker regional pattern.

In addition, when the traditional agriculture in the western Mediterranean is considered, the basic production of cereals were wheat and barley, and more specifically wheat. This was common practice beyond cultural, religious or political contexts. Bread is the basic food for most of the population during the Low Middle Age and Early Modern Epochs, as is indicated by historiography. It logically follows, then, that a permanent and complex monitoring system involving several public institutions (professionals, church, etc.) would focus on the most basic food production. Unfortunately, cultivation of other agricultural products does not have a similar monitoring system. To use other products for climatic proxies, we would have to collect information in an indirect way, such as from taxes statistics, tithes and other economic indicators. The reliability of the economic or tax information in Spain, however, is relatively low and irregular due to the usual practice of farmers to hide it. We do not include such information in the manuscript.

3) I have doubts on the classification of the rogations (lines 205-207, Table 2). Were the ceremonies the same in all the cities and during the whole time period, from 1650 to 1899?

Yes. From our direct reading of historical archives, we can state that institutional controls in pre-industrial society were so strict that many elements didn't change for centuries. We have added information in the discussion, lines 503 to 519.
*Nevertheless, the institutional controls in pre-industrial society were so strict that many of its constituent parts remained unchanged for centuries, and rogation ceremonies are one such element. This can be explained by two different factors. First, rogation ceremonies are used within the framework of the Roman Church Liturgy, so changes can only be defined and ordered by the Vatican authorities. If there is a will to change criteria affecting the substance of liturgical ceremonies, all involved institutions must record considerations, petitions and decisions in official documents from official meetings, supported by public notaries. In addition, changes must be motivated from the highest institutional level (Pope) to the regional authorities (Bishops) and local institutions (Chapters, parishes...). This system was too complex to favor changes. A second mechanism guarantees the stability of the rogation system: if any minor or important change in rogations is instigated at local level by the population or local institutions, this interference directly affected the Roman Church Liturgy. However, it was a change not to be taken lightly as the Inquisition Court would start judicial proceedings, and could bring a criminal charge of heresy. The punishment was so hard that neither institutions nor the people were interested in introducing changes in rogations.*

Severity indices are based on the type of ceremony, but is it a reliable criteria? In the discussion (lines 404-411), authors say that 'an index of level 2 does not necessarily imply that a drought was twice as intense as a drought classified as level 1, nor that the change in the intensity of droughts from level 1 to level 2 or from level 2 to level 3 has to be necessarily equivalent'. In that case, how must we interpret these indices? In my opinion, these indices only specify the nature of the ceremonies organized as response to natural hazards, but do not inform on the severity of the climatic event. In consequence, what is their utility from a climatic point of view? In my opinion, the binomial distribution (occurrence or not) is the more appropriate statistical approach to the treatment of this information.

The mechanisms underlying the triggering or absence of a rogation ceremony have been widely discussed in Barriendos et al., 1997; 2005 in Vicente-Serrano and Cuadrat, 2007 or even under the framework of a European funded project (Domínguez-Castro et al., 2012). The rogation ceremonies are indirect sources of climatic extreme events. The different types of rogations have a progressive meaning of climatic stress. For example, when a rogation of level 3 is performed, it means that the previous efforts of the society, municipal authorities and the church performing rogations of level 1 or 2 have not yielded results i.e. useful precipitation. Using a quantitative index such as the SPI, we could identify, for instance, droughts with an index of -2 and argued that those are twice as intense as those with an index of -1. However, here we cannot argue, unlike with other quantitative indices like the SPI, that a drought classified with level 2 is twice as intense as one with level 1. Yet, we can infer with high confidence that if there was a drought of level 2 it is because those types of ceremonies of level 1 did not work, and therefore the drought was still an issue for the development of the crops i.e., there is a progressive drying, but it does not have to be twice as intense. With this modified paragraph (lines 531-542) we tried to be honest with the semi-quantitative nature of our data and thus explain the potential drawbacks.

In addition, we are trying to evaluate the spatial scope of the rogations as a climate proxy. We also perform a comparative analysis of the local series in an attempt to construct regional series (in which the local binomial character is modified). For a local analysis, the occurrence may or may not be the most appropriate technique. For our attempt to regionalize, the approach proposed (and widely discussed) is justified. Finally, the transformation from categorical to semi quantitative values considers the incorporation of uncertainties, widely discussed in the text. Despite this potential loss of precision, our results show common and consistent patterns on a regional scale (Mediterranean and Ebro Valley). Not included in the manuscript.

4) Clustering is an appropriate tool to classify and group local series into regional series. There are very different clustering algorithms, hierarchical and not hierarchical. Why have you used Ward method with Euclidean distance, and not, for example, the non-hierarchical k-means, or other methods as the principal component analysis? Results of clustering must yield groups more or less homogeneous, but the chosen number of clusters is normally arbitrary. Why do you distinguish between Mediterranean and Ebro Valley group (dendrogram, Figure 3), if, as you say (lines 450-451) 'the high correlation between DIEV and DIMED is suggesting similar climatic characteristics'?

We know from observations that, although DIEV and DIMED locations have similar climatic characteristics, the Mediterranean coast locations have slightly higher precipitation totals. Then, we tried to differentiate that by developing two drought indices, which is supported by the cluster. One is reflecting the Ebro Valley conditions and the other is reflecting a more Mediterranean-like climate. Therefore, our final grouping is not only statistically significant, but it has also a geographical/physical meaning. Included in lines 428-429.

5) Validation of the regional drought indices is made using the overlapping period 1786-1899 between documentary and instrumental data. But, as you say in the discussion (lines 390-392) 'the apparent low frequency of rogations in the 19th century could be explained by a combination of political instability, and the loss of religiosity and historical documents'. I would add changes in the socioeconomic structures, organization of the cereal production, agricultural techniques, etc. In consequence, this period is not valid to calibrate and/or validate the rogation series in previous centuries. The cultural background, economic organization and technology of the 19th century was not the same that in previous centuries, and calibrations established for 19th century are not applicable to 17th century! In fact, you do not use this calibration (regression in Figure 5) to interpret previous data, only to validate the index during the 19th century.

We partially agree with this comment. It is true that there were changes in society and the political sphere but when there was an intense drought, the mechanism the society had to prevent it was still performing rogation ceremonies. That is also validated by the high correlation with the instrumental series. Additionally, the calibration period begins in 1787 when rogations were still deeply established in the society. We could show how internal historiographic criteria is applied to this manuscript, but we consider it is beyond the scope of the main message of the manuscript.

In any case, it is true that after 1834, when the Inquisition Court was abolished, liberal and democratic movements arrived from Europe and Spain experienced a strong process of anticlericalism, including a certain loss of religious values. This process obviously affected the occurrence of rogations and respective records on documentary series of public institutions. Still, this affirmation is valid only for specific areas or social contexts, because history is complex, non-linear and the development was not homogenous.

Besides, this analysis is only made for DIMED (Barcelona), and not for the other points in the studied area. I suggest to remove this analysis (and the Figure 5).

The analysis is performed for both DIMED and DI Barcelona included in DIMED, showing similar results and consistency through time. This analysis was a specific request by one of the reviewers, and we believe is highlighting the validity of the derived drought indices where we have instrumental data. Even with all the potential uncertainties, this comparison is one of the few forms of validation of this type of information given the absence of overlap with instrumental series. We prefer to keep it as is.

6) Superposed epoch analysis (SEA). Although you give a reference, a brief explanation on the basis and procedures of this method would be important.

Done. See lines 293-308.

7) Minor questions:
Line 301:'The cluster analysis (CA, see methods) using the DI of the 13 locations for the period of 1650-1899 AD revealed three significantly coherent areas…' Erratum, I suppose, clustering is made using the period 1650-1770, common to all the stations, although the classification obtained is after applied to the complete period until 1899.

Sorry for the misunderstanding. As indicated in the methods, the cluster is made using the period 1650-1770. We have corrected this (lines 328-330).

Lines 373-375: 'However, two years after the Tambora eruption in April 1815, there was a significant (p<0.05) increase in the three drought indices…' However? The time life of volcanic aerosols in the atmosphere is around one to two years. The Tambora aerosols caused a radiative forcing of the global climate system of about 5.6W/m2 for one to two years following the eruption (Brönnimann and Krämer, 2016). In consequence, this increase in the drought indices may be caused not by the volcanic eruption, but by the return to 'normal' conditions (or not forced climate variability).

The hydroclimatic volcanic response has been recently study using tree-ring records from Europe (Gao and Gao, 2017; Rao et al., 2017), showing a persistent drying in southern Europe two years after the volcanic event, which is consistent with our results. In addition, a previous study with rogation ceremonies (Dominguez-Castro et al., 2012), also highlights the year 1817 as one of the driest and it is attributed by the climatic effects induced by the Tambora event. All of which is considered in the results (lines 397-405) and discussion (lines 598-617) sections.

Lines 417-418: 'the local series are compared with the regional reference series as a basic element of quality control'. But, if the regional series is obtained from the average of local series, here we have a circularity problem.

We appreciate your concern. To solve it, we performed a modified version of the leave-one-out calibration method. Both the SPI and the SPEI are then correlated with a DIMED version that excludes an individual DI each time. The results are shown in the figure below (now Figure S1). We found significant correlations ($p<0.001$) on each case, including correlations above 0.45. We believe this example clearly shows that we do not have a circularity problem. We now include this in the methodology (lines 282-83) and in the results (367-369), and as Supplementary Figure 1.

[Figure]

Figure S1. A) 30y moving correlation between a DIMED version leaving one out and the instrumental computed SPI and SPEI. B) Same but 50y moving correlations. C) Correlation (Spearman) between DIMED leaving DIVIC out and SPIMAY_4, and SPEIMAY_4 for the full period (1787-1899). D) Correlation between DIMED leaving DITARRAGONA out and SPIMAY_4, and SPEIMAY_4 for the full period (1787-1899). E) Correlation between DIMED leaving DITORTOSA out and SPIMAY_4, and SPEIMAY_4 for the full period (1787-1899). F) Correlation between DIMED leaving DICERVERA out and SPIMAY_4, and SPEIMAY_4 for the full period (1787-1899).

Line 432: 'the local series are separated by tens or hundreds of kilometers'. If you speak on meteorological droughts, this is not a problem, because the dynamical conditions provoking dry conditions are associated to the predominance of anticyclonic conditions, and the spatial extension of an anticyclone may be much greater. Again, we are speaking on the drought concept. Meteorological, hydrological, agricultural?

We appreciate your comment. We clarified that we are indeed talking about a drought that links various characteristics of meteorological and hydrological drought to agricultural impacts. In any case, a recent study on the major drought episodes on Spain (González-Hidalgo et al., 2018) shows that there are some persistent droughts affecting large areas of Spanish mainland, including the mountain areas, associated to the predominance of anticyclonic conditions. However, in general, the spatial extent of droughts includes a great variability due to multiple factors (not included in the manuscript).

Figure 5. Significance level must be added in the figures. D), E), F), correlation is the Pearson correlation coefficient?

We appreciate the suggestions. However, we already included the significant level together with the Spearman correlation in the latest version of the manuscript. Again, we believe the reviewers did not get the latest version of the manuscript. We are sorry for that. It is the Spearman correlation as stated in the caption.

Figure 7. The legend is arbitrary, why do you distinguish between 2.1-2.6 and 2.7-3 DI values?

The legend was based in percentiles. Now we include it in the caption.

Reference
Brönnimann S, Krämer D. 2016. Tambora and the 'Year Without a Summer' of 1816. A perspective on Earth and Human Systems Science. Geographica Bernensia G90, 48 pp, doi:10.4480/GB2016.690.01

**Reviewer 4. Accepted subject to minor revisions.**
Rogation ceremonies: key to understand past drought variability in northeastern Spain since 1650. This study compiles and quantitatively analyses drought indices derived from documentary data on rogation ceremonies in northeastern Spain to further insights into historical droughts but also to understand the role of volcanic forcing on past event.
The most importantly contribution of the paper is that it provides a very interesting insight into both the strengths and weaknesses of documentary sources, especially approaches that seek to derive quantitative estimates from qualitative data. It is my view that the paper should be accepted following minor revision.

Many thanks for your positive comments and constructive review.

I suggest some points below that that the authors need to address. Most of my comments seek clarity and explanation. I also request the authors to give the paper a thorough edit. There are instances of misspelling or improper English scattered throughout the paper. This deserves considerable attention. There are also instances of long tracts of text that make it hard for the reader to track key points. Please use paragraphs more effectively to deal with this. There are also very long sentences at times (e.g. end of introduction) that need to be broken up.

Done.

I would like to know more about the documentary sources consulted. We don't get to know much about this aspect despite so much interpretation later depending on these sources. In line with comments from previous reviewers the original sources and their consultation/analysis needs to be given greater attention in the paper.

We provided detailed information about the documentary sources in the supplementary documents-Table S1. We have now moved that into Table 2.
In any case, a complete relation of documentary sources consulted requires an enormous display of pages of references. Case by case, each document can take hundreds of pages. A summary (as displayed now in Table 2) quantifying the collected sources is usually accepted. The main issue in displaying our effort is because we worked with primary sources, such as the administrative documentary sources, where every location usually contains 400-600 volumes of manuscripts. Rogations and other climatic descriptions are contained into these documentary series. Making a reference for every case would imply producing a book of references, which is not practical considering present editorial criteria. Users must be confident we work with all the quality requirements to preserve traceability of all the information used into the present manuscript.

The key methodological steps in the paper are as follows:
1)      Development of a semi-quantitative series from qualitative data derived from documentary sources on rogation ceremonies. This is done using an established technique. I have no concerns to note.

2)      Clustering of series to develop regional drought indices
This again seems to follow best practice. Importantly the analysis is not entirely statistically based and physical reasoning around the derived clusters is given. This is important as such techniques are somewhat subjective and the authors are transparent in their choices. The limitations of the approach are clearly articulated in the discussion.

3)      Validation of the resultant series against instrumental records.

4)      The performance of an epoch analysis to detect volcanic influence on historical droughts. This section is given the least attention in methods and most prominence in the abstract. I think that the authors need to explain this approach in more and sufficient detail to allow reader fully understand what they are doing here. A short paragraph should suffice. Why this method and what are the assumptions/strengths/weaknesses around the approach and desired attribution statements.

Many thanks for highlighting the merits of the manuscript. We now include a more detail section on the superposed epoch analysis (lines 294-308) and strengths/weaknesses are addressed in the discussion section.

I do not have local knowledge of the region but I find the results interesting, especially for the mountainous region. It seems the other two regions show similar results that are coherent with findings from previous studies. Indeed in discussion this aspect of the coherence of results needs to be moved further up. This is important information to have before getting into the limitations. Thanks for this suggestion, we now moved it further up.

I find the weaker results from the mountain region interesting. Some effort is expended on trying to explain why this difference and at times the authors get into attributing different processes. First, is this something seen in recent times when we have measurements?

This is an interesting point and the answer is not straightforward. First, the social and agrarian political system changed drastically during the first third of the 20th century with the construction of reservoirs. Both the Ebro Valley and much of the Catalan coast are flanked by two large mountain systems, the Pyrenees to the north, and the Iberian system from northwest to southeast. It is in these mountains where the highest rainfall occurs and therefore where the reservoirs are located. The ability to manage water resources became a determining factor for the local agriculture, which little by little began to generate more irrigated agricultural lands and less rain-fed agriculture. Second, in situations of anticyclonic stability that prevent the arrival of fronts loaded with humidity in the Ebro Valley, there are frequent convective storms in the Pyrenees and in the Iberian System. Although is true that most of the instrumental stations in Spain are located under 1,000 masl, we now know from observations that mountain regions are less affected by droughts than the lowlands. As noted above, there is a recent study on the major drought episodes on Spain (González-Hidalgo et al., 2018) showing that there are some persistent droughts affecting large areas of Spanish mainland, including the mountain areas. However, in general, the spatial extent of droughts includes a great variability (not included in the manuscript).

Second, given that the performance of a rogation was done based in part on the wishes of agricultural institutions, is there a risk that mountainous regions would have weaker political power in influencing rogations. Therefore, the lack of intense droughts in mountainous region or the disagreement with other regions, may be due to its weaker economic importance rather than anything to do with drought directly? I think the authors need to mention this possibility.

Your comment is very appropriated and well-focused. This argument could be negative for rogations in mountainous areas. But the rogation ceremonies system was not organized in a hierarchical structure, meaning that solicitudes were not sent to urban areas in lowlands, bishop authorities, or other administrations. Rogation ceremonies were not decided or prioritized from a distance. In fact, rogations ceremonies were adjusted to the capacities of every local community. This system is adapted to produce quick and efficient responses from local ecclesiastical authorities to local civil authorities, assessed by agricultural associations. The main idea of rogations is that after the first warning, the celebration of a rogation will be held within the next 4-6 days.

With the present manuscript and other works in progress, we are able to distinguish different climatic variabilities, confirming possible singularities in mountain areas. Rogation ceremonies help in this approach, because they can detect a range of climatic change from local sensitivities to large scale regional events or even global processes (such as volcanic events) (not included in the manuscript).

I also think that the authors could make more of the issues they run into for the mountainous region as a case of the challenges of using documentary sources. This needs to be mentioned in the abstract as its lessons are important for other studies.
Done.

The authors rightly state that the drought index for the mountainous region should be treated with extreme caution.
Thank you for your considerations. That is what we try to express in the revised version.

Need to be careful of tense used in abstract.
Done.

References
AEMET, 2019. State Meteorological Agency. Retrieved March, 11th 2019 from: http://www.aemet.es/es/serviciosclimaticos/datosclimatologicos/valoresclimatologicos?l=8368U&k=arn

[revised manuscript text omitted]

---

## Author Response (AR4)

*Dear editor, this is the point by point response with which we aim to respond all the comments and suggestions raised by the reviewers.*

**Referee #3**
**Rogations are a 'cultural' proxy. Therefore, they are affected by a certain degree of subjectivity, due to the perception of people about hydroclimate events. In consequence, the analysis must be cautious, taking into account their historical and sociological nature. My main criticism to studies based on rogations is that they often ignore this problem, perhaps because attempt to reconstruct long and continuous series of droughts (and floods).**

*We agree with this general comment. In fact, we tried to focus our manuscript not only in the elaboration of a reconstruction of past drought in the north east of the Iberian Peninsula using rogations, but mainly on a critical discussion on the potential and limitations of this proxy across different climate areas and different historical periods. The structure of our previous discussion section was perhaps confused. Thus, in the current version we included some additional paragraphs and reorganize the discussion to clearly highlight both the potential but especially the limitations of our findings.*

*The Historical/social character of rogations as a proxy is now stated early in the discussion section. Their degree of subjectivity is now clearly specified and their suitability in different historical periods is evaluated. We now highlight their limitations in the discussion section as follow i) its binomial character, ii) the unclear temporal scale at which they operate, iii) their spatial representativeness or iv) the impossibility in some cases to perform a real calibration/verification approach.*

*We hope this new version fulfill your expectations.*

**I suggested a seasonal study, including autumn months (rainfalls in October and November may be important for sowing), but it seems that authors have not considered this idea. Well, I can accept the definition of an annual index. However, the seasonal treatment would may shed light on the ordinal scale introduced (levels 0 to 3).**

*Due to the cumulative character of drought, the delays between drought and rogation occurrence and their differential influence on different agricultural species, techniques and environmental conditions (from coastal Mediterranean to mountain areas) an accurate definition of the temporal scale of drought that is represented by the rogation is challenging.*

*In this paper, for comparative purposes, a conservative approach is used. We combined rogations occurred from December to August in an index trying to account for general drought conditions occurred during the whole crop growing season across the whole study area (spring and summer) but also including previous conditions that may have impacted the final production (spring and winter rogations are likely to reflect drought conditions occurred in winter and previous autumn).*

*Specifically, we agree that the precipitation that occurred in the previous autumn may have impacted on the occurrence of rogation ceremonies. Still, due to the delay between climate and rogation occurrence we expect that such autumn conditions are reflected in the occurrence of winter rogations.*

*We now include such explanation in the current version of the discussion section.*

**Authors have solved some of my doubts, but, in my opinion, certain problems persist, in particular in relation to the calibration using the overlapping period 1786-1899. I can accept that in 1787 "rogations were still deeply established in the society", but authors say that historical process after 1834 "affected the occurrence of rogations and respective records on documentary series of public institutions". If this is the case, it would be more appropriate to use the period 1787-1834 for calibration purposes. If early instrumental series are not sufficient, the best reliability test is comparing the occurrence and severity of droughts with evidences from other independent documentary data or proxies.**

*This an important point that we try to solve/clarify in the current version of the ms. Since correlations between DI and instrumental SPI series are significant over the full 113-year analyzed period and also in the different 30 and 50-year subperiods considered, we decided to maintain the validation analyses as it is (using the full period 1789-1899).*

*Nevertheless, we now include some additional sentences to better describe how correlations vary through time, and how the agreement between instrumental data and DI differs in two different subperiods (better agreement between 1787-1830`s and decline thereafter). An explanation/ discussion about the potential causes of such changes is then included in the discussion section. Lines 357-362, and 481-510.*

*In line with then main general comment quoted by the reviewer, we think that now, both the potential but also the limitations of "rogations" as climatic proxies (in relation to the periods where they can be more or less reliable) can be better emphasized and discussed thereafter in the manuscript.*

**The interpretation of levels seems be influenced by the cumulative character of droughts. In that case, the 'annual' DI values obtained as the weighted average of the number of level 1, 2, and 3 rogations recorded, are misleading. It would be more appropriate to consider for each year the maximum level recorded, because minor levels are in some sense subsumed in a level 3 rogation.**

*There are different methodologies to deal with information derived from historical documentation including all of them different advantages and shortcomings in their interpretability (discussed in Gil-Guirado et al., 2019). Here again we adopted a conservative approach using a well-accepted approach.*

*Our goal is not only the development of a new drought index but also to test whether traditional approaches, used usually for local studies, can be generalized at regional scales.*
*Using only rogations of level 3 would be useful to compare extreme drought events but in such case, the influence of droughts of lower intensity will be neglected.*
*Defining whether a rogation of level 2 in a mountain area can be equivalent to a rogation of level 3 in a valley area is also a challenge due to social and historical differences between sites.*
*Again, for comparative purposes and despite the limitations, we chose for a conservative approach by integrating all available information in a single index, which despite its complexity may represents a general view on how drought operates across the region.*

**For instance, when authors affirm that "a dry winter can give any 'preventive' rogation of level 1 in early spring… if drought is persistent during the following spring, rogation ceremonies convoked by institutions are increased to higher levels". In their letter they say that "if there was a drought of level 2 it is because those types of ceremonies of level 1 did not work". Must we infer that a level 2 rogation was always preceded by a level 1 rogation?**

*The interpretation of the drought index has an intrinsic complexity due to the specific characteristics of the information contained in the historical documentation. We try to highlight such complexity in this paragraph. More specifically we modify the commented sentence by including "if occur". That means that a level 2 of rogations was "not necessarily" preceded by a rogation of level 1. Lines 458-462.*

**Finally, a comment on the hydroclimate responses to volcanic forcing. According to Fischer et al (2007) there is a tendency to anomalously dry conditions over the Iberian Peninsula in the winter of years 0 and 1 following volcanic eruptions, resembling a positive phase of the North Atlantic Oscillation. But the behaviour in spring is different, showing significant wetting over the western Mediterranean 2 years following an eruption (Rao et al., 2017). These authors explain that volcanic forcing may modulate spring and summer climate by stimulating a negative East Atlantic Pattern response. Therefore, we have a different behaviour in winter (dry, positive NAO) and spring (wet, negative EAP). This seasonal detail is not commented by authors**.

*We agree that the hydroclimatic and temperature response to large volcanic eruptions may vary depending on the season being analyzed and yet, Rao et al., 2017, as well as Gao and Gao 2017 suggest an intensification of drought conditions two years after volcanic eruptions in southern Europe. It should be noted, however, that these studies are based on the OWDA dataset, which consists entirely of tree-ring records and targets only June, July and August. In the analysis of Fischer et al., 2007 (of which one of the co-authors of this study is also co-author), they use a multiproxy dataset and although it also analyzes the conditions of the winter season, the number of proxies used for that season in the Iberian Peninsula is very scarce. However, in the analysis of Trigo et al., 2009 (also signed by two co-authors of this study) and specific for the Iberian Peninsula using the few instrumental series available highlights 'it seems however, that the winter of 1816/1817 and the following spring of 1817 were relatively dry in all three sectors of Iberia covered by these stations (although data from San Fernando was only available after January of*

*1817). In fact, based on the values from the three stations available, it is possible to state that the most important rainy season in Iberia (winter) was consistently dry between 1816 and 1819 in accordance with the results of the only work that had evaluated the impact of major tropical eruptions in the Iberian precipitation (Prohom and Bradley, 2002). It should be stressed that the precipitation total for 1817 in Barcelona was less than 200 mm (196.3 mm), roughly three times less than the long-term average value (573.7 mm) for the entire period with data (1786 - 1996), corresponding to the lowest value ever recorded in this city.*

They also analyzed some historical documents for different cities, as an example; '*This intense reduction (of precipitation) was observed throughout the whole year of 1817, without a particular focus on any season. In any case, the spatial configuration of this drought is variable in time, a fact that might be partially related to the large orographic complexity of Iberia and the corresponding large spatial gradients of precipitation characteristics (e.g. Rodriıguez-Puebl aet al.,1998; Serran oet al., 1999). Contemporary accounts describe the most severe examples of the problems caused by the drought namely the loss of cereal crops, the shortage and the high prices reached for many essential products (e.g. bread, milk, vegetables). At the end of 1817, the situation was particularly bad in many cities and villages. For example, of the 30 wells that normally supplied water to the towns-people of Arenys de Munt (40 km NE of Barcelona) only 6 had water, moreover this water was turbid and of poor quality. Naturally, the hydraulic energy obtained in water-mills was also severely reduced. All the watermills in the area were left dry and what little flour there was had tobe produced in Girona (60 km away) or at two emergency mills (so-called 'blood-mills') driven by people and horses (Archive of Arenys de Mar, Mem`ories de lacasa Belsolell de la Torre, 1816, p. 99).*

These are just some evidences in agreement with our findings (Figure 8b), and that support our hypothesis of overall significant wet conditions the year of the tropical volcanic event and one year after, follow by drier conditions 2 years after the event.
In any case and being aware that the climate response to volcanic forcings may vary according to the season, we have now included a sentence on the discussion (lines 634-637).

**Referee #4**
**I find that the authors have done an excellent job in addressing my points. The paper is an interesting and important contribution that shows the value of and challenges of using documentary sources – in this case information from rogation ceremonies. The paper now reads well and I very much like the discussion. It will be an informative resource for future work, and I commend the authors for so fully discussing the pros and cons of their work. I recommend accept with some very minor points below that I picked up upon reading.**

*Many thanks for your positive and constructive comments and suggestions that helped enhance the quality of the manuscript.*

**The main one is that the abstract could be shortened a little, there is perhaps too much detail in there that distracts from the key results. The final conclusion paragraph could also be strengthened by removing the first sentence.**
*Done.*

**Line 26, standardized precipitation**
*Done*
**Shorten abstract a little.**
*Done*
**Line 408 – in this paper**
*The discussion has been reorganized.*
***Line 430, don't start with Finally, instead start with We found…***
*Done. Now line 542.*
**Throughout – drought indices rather than Drought indices**
*Done.*
**Line 539, it is because**
Done.
**Line 549, in this paper**
Done.
**Lines 647 – 650 – I would recommend delete this sentence and start with 'Finally, the historical data from rogations….**
Done.

Gil-Guirado, S., Gómez-Navarro, J. J., and Montávez, J. P.: The weather behind the words. New methodologies for integrated hydrometeorological reconstruction through documentary sources, Clim. Past Discuss., https://doi.org/10.5194/cp-2019-1, in review, 2019.

[revised manuscript text omitted]